

**Hydrological processes and permafrost regulate magnitude, source and chemical characteristics of dissolved organic carbon export in a peatland catchment of northeastern China**

Yuedong Guo[1] , Changchun Song [1,*], Wenwen Tan[1], Xianwei Wang[1], Yongzheng Lu [1]

[1]Key Laboratory of Wetland Ecology and Environment, Northeast Institute of Geography and Agroecology, Chinese Academy of Sciences, Changchun 130012, China

Tel: 86-431-85542211

Fax: 86-431-85542298

Address:

   Northeast institute of Geography and Agroecology, Chinese Academy of Sciences.

No.4888, Shengbei Road, Changchun, Jilin Province, China, 086-130102

**Abstract**

   Permafrost thawing in peatland has the potential to alter the catchment export of dissolved organic carbon (DOC), thus influencing carbon cycling in linked aquatic and ocean ecosystems. However, peatland along the southern margins of Eurasian permafrost are seldom examined in spite of the presence of considerable risks associated with degradation due to climate warming. This study examines dynamics of DOC export from a permafrost peatland catchment located in northeastern China





during the growing seasons of 2012 to 2014. Our findings show that runoff processes affect observed DOC concentrations, magnitudes, sources, and chemical characteristics of stream discharge. The entire catchment exhibits strong potential for annual DOC exporting (4.87 g C m$^{-2}$), and DOC from the peatland landscape alone is estimated to amount to 12.89 g C m$^{-2}$. Annual DOC export processes are closely related to total discharge levels, and floods contribute to approximately 85% of DOC export levels. Flood volumes derived mainly from peat pore water stored in the upper organic layer of the soil profile prior to rainfall events, creating a strong linkage between discharge and DOC concentrations. DOC source and chemical characteristics, as indicated by three fluorescence indexes, have changed regularly according to source shifts occurring as a result of flood and baseflow processes. A deepening of the active layer due to climate warming should elevate proportions of microbial-originated DOC in the baseflow. Given expected future increases in precipitation, our results show that the magnitude of DOC exports from the study region will increase.

## 1. Introduction

Permafrost soils have acted as sinks for atmospheric carbon (C) since at least the late Pleistocene and serve as key sources of dissolved organic carbon (DOC) for linked aquatic and ocean ecosystems (Opsahl et al.,





1999; Kicklighter et al., 2013). As changes in the quantity and quality of exported DOC could greatly alter the energy cycles of linked oceans, considerable advances have been made in recent years to better evaluate potential changes DOC export patterns from permafrost regions (Townsend-Small et al., 2011; Vonk et al., 2013;). However, uncertainties remain regarding to main driving factors involved and the fate of DOC due to complex interactions between hydrological and thermal dynamics and bio-chemical drivers (Olefeldt and Roulet, 2012; Kicklighter et al., 2013).

Significant losses of near-surface permafrost have been observed over the past century and such outcomes have induced considerable changes in hydrological processes and soil thermal regimes (Lyon et al., 2009; Lessels et al., 2015), in turn altering the magnitude and timing of terrestrial DOC export processes. Hydrological processes are an important and well-documented regulator of DOC export from permafrost regions (Ågren et al., 2010; Guo et al., 2015). Owing to increased levels of hydrological access to previously frozen soils following permafrost degradation, DOC export is forecasted to increase in Siberian rivers along a latitudinal transect (Frey and MacClelland, 2009). However, permafrost degradation also increases the likelihood of interactions occurring between subsurface flows and mineral soils, which in turn lead to considerable levels of DOC absorption by fine soil particles and which



decrease levels of DOC export magnitude (Petrone et al., 2006; Striegl et
al., 2007). There are significant disparities in DOC export concentrations
and seasonal patterns between surface- and subsurface-dominated runoff
processes in permafrost catchments (Laudon et al., 2011). Studies have
proven that capacities for DOC export from permafrost soils are closely
related to lateral subsurface flows (Striegl et al., 2007; Lyon et al., 2010).
Therefore, alterations in hydrological pathways during permafrost
freeze-thaw cycles are some of the most important factors to consider in
evaluating DOC export potential.
Hydrological pathways control not only on the availability of DOC
produced along the surfaces of soil particles but also hydrological
connectivity at the landscape scale, which determines physical transport
routes of DOC release from a catchment (Birkel et al., 2014). A strong
hydrological connection generally contributes to high levels of DOC
export magnitude (Olfeldt and Roulet, 2012; Lessels et al., 2015). A
catchment with 4% peatland cover achieve a 12% higher level of annual
DOC export than an upland catchment due to maintained levels of
hydrological connectivity (Olfeldt and Roulet, 2014). However,
uncertainties remain in predicting DOC export processes based on
changing hydrological processes. Levels of annual DOC export from
permafrost are known to vary greatly between different landscapes and
regions, e.g., at rates of between 1 and 35 g C m$^{-1}$ yr$^{-1}$ (Fraser et al., 2001;



Dinsmore et al., 2010), where variations mainly relate to hydrological
regimes (Holden, 2005).
Flow pathways also determine chemical compositions of DOC
export from permafrost catchments, which in turn have considerable
impacts on downstream DOC mineralization levels and on C emissions
from streams, lakes and oceans (Mann et al., 2012; Cory et al., 2014).
Runoff processes can alter export compositions to a certain degree
according to pathways of the organic-mineral soil layer. Mineral soil
particles can stably absorb dissolved organic matter high in aromatic
components with large molecular weights or acidic functional groups
(Kalbitz et al., 2005) as well as aromatic structures while hydrophilic
fatty microbial products with low molecular weights are desorbed and
released (Striegl et al., 2005). To date, no satisfactory theoretical
framework or method has been developed to quantify alterations in DOC
chemical characteristics following permafrost degradation. More detailed
surveys on comprehensive effects of hydrological processes are still
needed to predict alterations in the magnitudes and chemical
characteristics of DOC exported from permafrost catchments.
Given the high spatial heterogeneities of peatland and complexities of
hydrological processes in permafrost regions, it is important to
understand magnitudes and regulations on DOC export in different
permafrost regions and especially in the south part of the Eurasian





continent where limited research has been performed. This study focuses
on dynamics of DOC release from the Fukuqi River, a tributary of the
Amur River positioned along northern slopes of the Great Xing'an
Mountains in northeastern China. The Great Xing'an Mountains form an
important barrier from Siberian cold air masses and monsoons of East
Asia. The mean annual temperature of the area has on average increased
by 0.3 ℃ every 10 years over the last 50 years, subjecting permafrost to
higher risks of degradation. On the southern slopes of the Great Xing'an
Mountains, the thickness of the active layer has increased by 20-40 cm
from 1970s to 2000 (Jin et al., 2000). However, few studies have focused
on possible effects of permafrost degradation on this region to date. This
work thus investigates potential changes in DOC export patterns by
answering the following questions:
(1) How much DOC is exported via stream discharge during the
growing period?
(2) What is the relationship between runoff processes and
concentrations, sources, and chemical characteristics of DOC?
(3) What are the potential effects of permafrost degradation and
climate change?

**2.   Approach and methodology**
**2.1. Study area**



Northern sections of the Great Xing'an Mountains in China are
located along the southern margins of the continuous permafrost zone in
Eurasia. The area represents the most remote region of the East Asia
monsoon of the East Eurasian continent. The region includes
approximately $8.245 \times 10^3$ km$^2$ of natural wetland, representing a major
proportion cold temperate wetlands and an important reservoir of soil
carbon and usable water resources for northeastern China.
The Fukuqi River, a second order branch of the Amur River, is
located at continuous permafrost zones of the northern section of the
Great Xing'an Mountains (Fig. 1). The catchment extends across an area
of 286.86 km$^2$ with an annual mean temperature of −4.2 °C and a mean
annual precipitation level of 425 mm (1959-2013). Large peatland areas
have formed throughout the flat river valley. The peat layer, which is
approximately 0.3-0.4 m thick, is composed of typical organic soil with
organic matter levels ranging from 40% to 60% and with porosity levels
ranging from 60% to 20% from the surface. According to previous field
survey, the peatlands accounts for more than 90% of the total carbon
stock in the catchment although it covers only about one-third of the total
area. The maximum thaw depth of the active layer, ranging from 60 to 80
cm, occurs usually in early August. Below the peat soil layer, there
covers mineral soil with much lower organic content (< 5%) and soil
porosity (< 10%) than the upper soil. The plants usually grow from May



until late September. The Sphagmum mosses (*S.capillifolium, S.*
*magellanicum*) and sedges (*Eriophorum vaginatum*) are the dominant
vegetation. The upland mountains on both sides of the valley are
extensively covered by mineral soil and gravels with little organic
content due to the continuous logging and frequent fires during the past
60 years. To date, the original coniferous forest has been already replaced
by yang *Pinus sylvestris var. mongolica*.

**Fig. 1** Geographic location of the study area.

**2.2. Sampling and monitoring program**
Monitoring was conducted from early May to late September of
2012, 2013 and 2014. A gauging profile for DOC concentrations and
hydrological parameters was set for the lower reaches of the Fukuqi
River (Fig. 1). Water samples were collected from the stream profile
every 1–5 days, and a higher sampling frequency was applied during
flood periods while a lower sampling frequency was applied during low
water periods. A 200 ml clean polyethylene bottle was used to obtain
triplicate samples from the surface, middle, and bottom layers along the
vertical direction of the profile. The collected water samples were filtered
through a 0.45-µm glass fibre membrane. Then, DOC concentrations in
the samples were measured using a DOC analyser (C-VCPH, Shimadzu,


Japan) as soon as possible.
Meanwhile, the water level and mean flow velocity in the profile
was automatically measured to evaluate stream discharge (Q) by a water
level monitor (Odyssey, New Zealand, accuracy: ±2 mm) and a flow
meter (Argonaut-ADV, USA, accuracy: ±0.01 m/s) respectively. Air
temperature and soil temperature at 0–1.0 m depth were also recorded by
an automatic microclimate gauging tower (Campbell, USA) set in the
center part of the peatlands. The standing water levels were successively
recorded by the same Odyssey monitor in the site nearby gauging tower.
The thaw depth of the peatland active layer was manually surveyed
weekly with a 1.0-m stainless steel ruler (accuracy: 0.1 cm) at the same
three sites. Information of the temperature (°C), electrical conductivity
(mS/cm), and turbidity (NTU) in the sampling profile is automatically
obtained by a multi-parameter water quality sonde (YSI6600, USA). Part
of the water quality data was lost in 2012 and 2013 due to the excessively
low temperature in the stream.

To assess chemical DOC characteristics of the active layer and to
determine sources of DOC in the discharge, peat soil pore water was
collected from three sites located 50-100 m away from the main river
channel in the growing seasons of 2013 and 2014 (Fig. 1). For each site,
3-5 sample points were used repeatedly for each sampling procedure.





When sampling, 100 ml samples of soil pore water drawn from different
depths at 10 cm intervals in the active layer were collected using ceramic
soil pore water samplers (SIC20, Germany). Due to the gradual thawing
of the active layer throughout the growing season, maximum sampling
depths varied. Meanwhile, rainfall samples were during the two growing
seasons. We ensured that the sample bottles did not contain any air in
adherence with analysis requirements for stable oxygen isotopes ($\delta^{18}O$‰).
The depletion of stable oxygen isotopes ($\delta 18O$‰) for the discharge
samples, peat soil pore water and local rainfall in 2013 and 2014 were
analysed with an isotope mass spectrometer (Finnigan Delta plus XP,
USA) at the Key Laboratory of Wetland Ecology and Environment,
Chinese Academy of Sciences.

**2.3. Fluorescence measurements**
Excitation-emission matrixes (EEMs) of the water samples were
measured using a Hitachi F-7000 fluorescence spectrometer (Hitachi
High Technologies, Japan) with a 50 W ozone-free Xenon arc lamp and
R928P photomultiplier tube fitted as a detector. The spectrometer was set
to collect signals using a 5-nm bandpass on excitation and emission
monochromators at a canning speed of 3,200 nm/min. EEMs were
recorded for excitation spectra of between 220 and 400 nm and for
emission spectra of between 300 and 500 nm. To eliminate the inner-filter



effect, samples were diluted with deionized water to a decadal UV
absorbance at $\lambda$= 254 nm of 0.2 absorbance units (cm-1). Milli-Q water
blank EEMs were subtracted from the sample EEMs to eliminated Raman
scatter peaks. Then, the EEMs were normalized to the area under the
Raman scatter peak (excitation wavelength of 350 nm) of a Milli-Q water
sample run the same day. The fluorescence intensities measured were
reported in Raman Units (RU) in this study.
Three spectral indexes calculated from the EEMs were measured to
quantify chemical characteristics of the dissolved organic matter: 1)
humification (HIX) defined as the ratio of the sum of $\lambda$em = 435–480 nm
to the sum of $\lambda$em = 300–345 for excitation at 254 nm and quantifying
the complexity and aromaticity of dissolved organic matter. High HIX
values denote the presence of highly humified or more complex organic
matter (Ohno, 2002); 2) the fluorescence (FI) defined as the ratio of
maximum emission fluorescence intensities at 450 and 500 nm for
excitation at 370 nm identifies sources of humic-containing dissolved
organic matter. The recommended FI for terrestrial-origin humics is 1.2
and that for materials of microbial origin is 1.7 (Cory et al., 2010); 3) the
biological index (BIX), defined as the ratio of intensities at $\lambda$em 380 nm
and 430 nm for excitation at 310 nm, is a complementary index for
evaluating the relative contributions of microbial-derived organic matter.
BIX values of 1.0 or greater correspond to freshly produced DOC of




microbial origin, whereas values of 0.6 and lower imply the presence of
little natural biological material (Huguet et al., 2009).

**2.4. Statistical analyses**
The mean and the standard deviation of the DOC concentration in the
stream and soil pore water, and the three fluorescence indices were
statistically analyzed with the Statistical Program for Social Sciences
(SPSS) version 13.0 software. The relationship between the hydrological
factors and the DOC concentration and the fluorescence indices was
examined by a two-tailed Pearson correlation and regression analysis,
where the p-values were calculated to test for significance.

**3. Results**
**3.1. Environmental conditions**
Substantial inter-annual and seasonal variations in precipitation were
observed for the three years (Fig. 2). Total precipitation levels reached
202.5, 520.8 and 164 mm in 2012, 2013 and 2014, respectively. Based on
our statistics on the regional climate dataset for 1970 to 2005, 2013 was
an extremely wet year due to excessive rainfall occurring in the spring
and summer. Precipitation levels in the growing season of 2012 remained
within a normal range while those for 2014 denote the presence of very
dry conditions in the study area. Influenced by unusual precipitation



patterns, the mean air temperature of the growing season of 2013, 12.9℃,
was somewhat lower than those of 2012 and 2014 (13.65 and 13.67℃).
However, all mean values fell within the average range for the long-term
climate dataset. We also found no significant differences in maximum
thaw depths for the three years, finding values of approximately 70 cm.
Standing water levels close to the stream channel declined overall across
the growing seasons. No recorded data for higher than peat ground
surface were detected for the three years.

**Fig. 2** Dynamics of air temperature, precipitation, standing water levels,
and thaw depth observed during the growing seasons of 2012 to 2014.

**3.2. DOC concentrations and fluxes**

DOC concentrations in the Fukuqi River fluctuated considerably with

discharge levels during the three growing seasons (Fig. 3). A maximum
concentration of 44.71 mg L$^{-1}$ was found for the early spring of 2013
accompanied by maximum flood levels for the three years. In the autumn,
when flows were relatively low, DOC concentrations generally fell below
8 mg L$^{-1}$. It is worth noting that consistently high concentrations were
recorded during two flood periods of the autumn of 2012 and of the
spring of 2013. A significantly positive correlation was found between
DOC concentrations and discharge levels for all three growing seasons



(n=92, p <0.01). Meanwhile, DOC concentrations were positively related
to discharge turbidity and negatively related to discharge conductivity
(n=68, p < 0.01) while no significant relationship was found between
concentrations and air temperature or for soil temperatures of the active
layer.
Mean DOC concentrations measured during the growing seasons
were measured as 13.84, 19.98 and 13.82 mg/L for 2012, 2013 and 2014,
respectively, with an overall mean value of 15.94 mg/L. Total DOC
export magnitudes for the entire catchment were estimated as 1055.71,
2467.37 and 672.59 t for the three respective years, denoting levels of
DOC export of 3.68, 8.6 and 2.34 g m$^{-2}$, respectively. Statistically
speaking, the nine flood events (maximum discharge $> 1.0 \times 10^6$ m$^3$ d$^{-1}$)
were responsible for 81% of the total DOC flux while the five floods
with a discharge level of $> 2.0 \times 10^6$ m$^3$ d$^{-1}$ accounted for 65% of the total
flux. In total, approximately 85% of DOC was exported during flood
periods.

**Fig. 3** Dynamics of dissolved organic carbon (DOC) concentrations and
discharge observed during the growing seasons of 2012 to 2014. The
discharge (Q) unit used is $10^6$ m$^3$ d$^{-1}$.

**3.3. Spectral indexes of DOC**



The three spectral indexes varied considerably in terms of discharge
processes during the growing seasons as is shown in Fig. 4. We found a
positive correlation between the HIX and logarithmic discharge whereas
both FI and BIX exhibited a significantly negative correlation with
logarithmic discharge (Fig. 5). HIX ranged from 5.52 to 16.41 with an
average value of 10.38, revealing a high volume of humification
components in the stream discharge DOC (Ohno, 2002). This index and
all of the other variables show significant relationships with hydrological
DOC, Q, conductivity, and turbidity (Table 1). FI and BIX values ranged
from 1.43 to 1.62 and from 0.46 to 0.63 with average values of 1.52 and
0.54, respectively. The FI values indicate that DOC was derived from
both terrestrial and microbial sources (Cory et al., 2010) while the BIX
value denotes the presence of a low volume of fresh organic matter from
biological sources in the runoff (Huguet et al., 2009). FI and BIX values
were only closely related to hydrological variables; no relationship to
temperature was found.

**Fig. 4** Relationships between discharge and the three spectral indexes
during the growing seasons.

**Fig. 5** Relationships between discharge and the three indexes during the
study period.






**Table 1.** Correlation analysis of the three fluorescence indices with hydrological and climatic factors.


### 3.4. Stable oxygen isotopes in rivers, soil pore water, and rainfall

We found nearly no seasonal variations in $\delta^{18}$O‰ values from the rainfall and soil pore water samples (Fig. 6). It seems that air temperatures and rainfall quantities had no effect on the depletion of oxygen isotopes in rainfall during the growing seasons. The mean value of rainfall was measured at roughly -7.62 ± 0.53‰, which is a significantly higher value than that found for river discharge and soil pore water ($P$<0.01). Two samples of river discharge collected in the early spring of 2013 clearly show higher values than those of the other samples, which fluctuated slightly around a mean value of roughly -14.64 ± 0.87‰ during the other period. The $\delta^{18}$O‰ values of soil pore water at the three sample sites did not vary by location or season. The mean value for the samples was recorded as -14.67 ± 0.49‰, which is not statistical different from the value found for the river discharge samples ($P$ <0.01).

**Fig. 6** Dynamics of stable oxygen isotope values for rainfall, discharge and soil pore water in the catchment.



### 3.5. DOC concentrations and fluorescence indexes of soil water

During the growing seasons, DOC concentrations in the soil pore water changed considerably with depth. Maximum DOC concentrations were typically found in the plant root layer (36.98 mg L$^{-1}$) while a minimum value of 15.36 mg L$^{-1}$ was found at the bottom of the profile (Fig. 7). DOC concentrations at different depths change to varying degrees during the growing seasons. However, we found no significant difference between average concentrations. There is a strong relationship between DOC concentrations and total soil organic matter, total nitrogen content levels and soil bulk density levels along the profile, and we found no relationships with soil temperature (Table 2). Similarly, no significant relationship was found between the average DOC concentration and the average soil temperature during the growing seasons.

**Fig. 7** DOC concentrations in soil pore water along the soil profile for 2013.

**Table 2.** Results of the correlation analysis of dissolved organic carbon (DOC) in the soil pore water with soil factors

The HIX, FI, and BIX of the soil water samples varied greatly with soil depth (Figure 8). We found a pronounced change in the three indexes





at 30-40 cm, where we found soil organic matter levels to suddenly
decline. HIX levels gradually decreased from the top to the bottom while
FI and BIX levels were found to follow the opposite trend. For all of the
collected samples, HIX levels were found to be significantly and
positively related to DOC concentrations and to soil organic matter
content levels (n = 18, p < 0.01) while FI and BIX levels were found to
be inversely and significantly correlated with those parameters (n = 18, p
< 0.01).

**Fig. 8** Vertical distribution of the three spectral indexes for soil pore
water along the soil profile for 2013.

**4. Discussion**
**4.1. Flow pathway and DOC concentrations**
DOC concentrations in permafrost catchments have been reported to
vary considerably, and flow pathways are the most influential controllers
of runoff events and entire growing seasons (Hagedorn et al., 2000;
Dawson et al., 2008; Guo et al., 2015). Peatland in permafrost generally
experiences subsurface flows but not over surface flows due to the
occurrence of high levels of rainfall infiltration into the thawed organic
layer (Carey and Woo, 1997). Resting on a frozen soil layer, infiltrated
and previously stored water is prevented from draining deeper, and a


lateral subsurface flow parallel to the bottom of the active layer forms. It
is worth noting that water previously stored close to the stream channel
typically forms a major proportion of flood peaks owing to the high
hydraulic conductivity of macroporous organic soil in peatland (Carey
and Woo, 2001).
In the Fukuqi catchment, the porosity of peat in the upper 40 cm layer
can generally reach levels of 20-60%. High infiltration rates and
hydraulic conductivity levels found in the organic soil layer enable flow
water to respond quickly to hydrological inputs and to transfer discharge
into the channel. This prevents the formation of overland or surface flows.
This serves as direct proof that the standing water level has never
exceeded the peat surface and even during two large spring floods
occurring in 2013. It is noteworthy that $\delta^{18}O‰$ values in the discharge are
generally similar to those in soil pore water close to the stream channel
while being more negative than those in rainfall. This shows that the
stream discharge is mainly composed of soil pore water reserved in the
peatland area before new rainfall events occur, proving that lateral
subsurface flows constitute the main form of runoff generation in the
catchment.
Lateral subsurface flows are a fundamental condition of the positive
relationship between runoff and DOC concentrations (Quinton and Gray,
2003; Birkel et al., 2014; Guo et al., 2015). Subsurface flows guarantee





that the soil pore water reserved in peatlands before the rainfall event was
pushed into the channel in order to the distance to the stream. The
preferential output of peat water close to the channel characterized by
high DOC concentrations contributes to a concentration peak occurring in
flood periods and thus to a positive relationship between runoff and DOC
concentrations during flooding periods, which lead to the same relation
for runoff processes.

**4.2. Hydrological connectivity and DOC export potential**
DOC exports from permafrost are also dependent on the connectivity
of flow pathways that mobilize and transport DOC to streams (Köhler et
al., 2002; Laudon et al., 2011). In permafrost catchments covered with
peat-dominated soils, geomorphic landscape structures have been deemed
crucial in determining the hydrological connectivity of peatland areas and
upland hill slopes (Dawson et al., 2008). We found that peatland
distributes along both sides of the stream channel as is shown in Fig. 1,
revealing hydrological connectivity between the peatland area and stream
during both flood and baseflow periods. It is noteworthy that the
two-sided distribution of peatland prevents runoff flows from marginal
hills from reaching the river channel directly despite entering the peatland
area first (Guo et al., 2016). This is likely why $\delta^{18}O$‰ values in the
discharge samples are similar to those of the soil pore water while not



being affected by water volumes from upland hill slopes, for which $\delta^{18}O$‰
values should be similar to those of rainfall. Therefore, water volumes
from upland areas, which contain low levels of DOC due to being
covered with mineral soil and gravel, should not dilute DOC
concentrations during floods. Simultaneously, permafrost across the
whole catchment plays a central role by blocking the input path of
shallow groundwater from upland areas. DOC concentrations during
floods are only related to those of peat pore water. Hydrological
connectivity maintained in the stream-peatland continuum centrally
supports high DOC concentrations observed during floods.
It can therefore be speculated that exported DOC in the catchment is
mainly "autochthonous" and derived from riparian peat throughout the
growing season contrary to the recently accepted view that the source of
DOC realised from headwater catchments is "allochthonous" from upland
soils at least during wet seasons (Boyer et al., 1996; Inamdar et al., 2006;
Sanderman et al., 2009). In fact, "autochthonous" DOC export results not
only from the simple landscape structure of the stream-peatland-upland
continuum along the catchment transect but also from the high DOC
production capacities of peatland. Though we did not conduct direct
experiments on this issue, the field data show high levels of DOC
production potential for peatland in the catchment. According to our data
on soil pore water, DOC concentrations in the peat pore water have



always been high across seasons and were accompanied by several large
floods in 2013 (Fig. 7). Importantly, DOC concentrations in the discharge
show no clear drawdown during two successive flooding periods in the
spring (Fig. 3), revealing the weak influence of successive exports of
floodwater on DOC concentrations in soil pore water. As a balance
between DOC production and dissolution can re-occur within hours under
suitable conditions (Worrall et al., 2008), the DOC production rate is not
the limiting factor that controls export concentrations.

### 4.3. DOC sources and chemical characteristics

Carey and Woo (2001) describe permafrost soil in reference to a
two-layer flow system based on the difference in hydraulic conductivity
between the upper organic soil layer and lower mineral soil layer. As
hydraulic conductivity levels typically decline exponentially in the
transition from organic to mineral soil, DOC levels in discharge during
the flood period derive mostly from the upper organic layer while DOC
levels of the recession and baseflow periods are mainly derived from the
lower mineral soil layer in the study catchment (Guo et al., 2015).
Therefore, considerable variations in DOC chemical compositions along
the vertical soil profile are bound to affect their performance in the
discharge examined in this study. The three fluorescence indexes of HIX,
FI, and BIX, which generally show no changes for diluted DOC



concentrations, serve as robust indicators of sources of stream discharge.
Considerable variations in the three indexes observed during flood events
prove the occurrence of changes in DOC sources following a shift from
the flood to the baseflow phase. Flood DOC, which is characterized by
higher levels of humification and by the presence of few
microbial-originated organic components, can be identified from the
upper soil layer of the peatland area while DOC in the baseflow is
derived from the lower mineral soil layer. Previous studies of permafrost
catchments have recorded shifts in DOC compositions attributed to
different source water contributions across seasons (Spencer et al., 2008;
O'Donnell et al., 2010). However, our results highlight shifts in DOC
chemistry through rainfall-runoff processes. Lambert et al. (2014) have
also confirmed the presence of a shift in DOC sources between riparian
wetland areas and upland areas during flood events in a DOC-limited
upland catchment, revealing the unpredictable complexity of DOC source
during floods due to effects of landscape structures, hydrological
pathways, and organic carbon mineralization patterns. From our study
results, it is easy to understand that the shift in water sources from the
upper soil layer to the lower soil layer during flooding can be attributed to
great discrepancies in hydraulic conductivity observed in the autumn
once the lower mineral soil layer has thawed. However, such shifts are
also clearly observed in the spring when the upper soil is still frozen (Fig.





2 and 4). This suggests the presence of another DOC source such as litter
covering the peatland area in the spring, which could easily release DOC
from rainfall extraction. Thus, a more detailed survey of litter- and
upland-originated DOC is urgently needed in future research.
The deepening of the soil active layer will alter the discharge flow
pathway and in turn DOC sources and chemical characteristics. As few
rainfall events occurred in 2014, we were able to identify effects of the
gradual deepening of the active layer throughout growing seasons. We
find a remarkable elevation in BIX and FI levels in discharge from the
spring to the autumn when flood periods are disregarded (Fig. 4),
highlighting the influence of active layer depths on DOC sources and
chemical characteristics. The elevations found suggest that an increase in
microbial-originated DOC from the lower soil layer increases discharge
levels following the deepening of the soil active layer. This result is
consist with the conclusions of Prokushkin et al. (2007) who also found
higher levels of microbially transformed and/or derived material export
due to the presence of a deeper active layer in the summer and autumn in
Siberia. Changes in biochemical compositions (decreases in the
lignocellulose complex; increases in the hydrophilic fraction) are
confirmed further in Kawahigashi et al. (2004).
From our results, the humification degree of DOC as determined by
HIX shows no clear trend for the seasons of 2014. HIX values fluctuate



considerably even following a minor flood, showing that the hydrological
process considerably controls the humification of exported DOC. As is
shown in Fig. 8, HIX values in the deeper soil layer change little from
June to August while BIX and FI values do not, and this likely spurred
the differing performance of the three indexes in terms of stream
discharge levels during the seasons of 2014. It can be concluded that FI
and BIX values are both sensitive to flooding processes and soil active
layer depths while HIX values only respond to flooding processes.
Therefore, the three indexes respond differently to different
environmental factors, and a joint analysis will help reveal the chemical
characteristics of organic matter synthetically.

**4.4. Export magnitude and potential**
Several characteristics of permafrost peatland (e.g., high organic
matter content levels, low water temperatures, weak microbial
transformations, and high levels of hydraulic transmission) result in large
magnitudes and in strong potential for DOC export (Balcarczyl, et al.,
2009; Lessels et al., 2015). According to our data, levels of net DOC
export from the studied catchment are estimated at 4.87 g m$^{-2}$ yr$^{-1}$, which
is in the lower range of reported permafrost estimates ranging from 1 to
35 g C m$^{-1}$ yr$^{-1}$ (Fraser et al., 2001; Dinsmore et al., 2010). Roughly
two-thirds of the catchment is covered in mountain gravel and mineral



soil with low levels of organic carbon in the Fukuqi catchment, likely
decreasing the mean DOC export capacity of the whole catchment.
According to our field survey, carbon stock in the peatland area accounts
for approximately 90% of carbon levels in the catchment (Miao, 2014).
Assuming that the DOC originates from both peatland and forests and
that the total export level is proportional to the organic carbon pool in the
soil of each ecosystem, DOC exported from the peatland area can be
estimated at 12.89 g m$^{-2}$ yr$^{-1}$ on average. According to Miao (2014), the
net ecosystem exchange (NEE) of peatland in the study catchment
determined from carbon dioxide and methane fluxes between peatland
surfaces and the atmosphere is $30.59 \pm 1.98$ g C m$^{-2}$ yr$^{-1}$. Therefore, the
DOC export magnitude should account for 72.8% of the NEE of the
peatland area, as the verified NEE was calculated as 17.7 g C m$^{-2}$ yr$^{-1}$
(30.59-12.89). Theoretically, the data are overestimated due to our
assumption of a linear relationship between carbon storage and DOC
export magnitudes. However, our data still highlight the importance of
stream carbon export for peatland net ecosystem carbon balance. Any
disturbance altering DOC export magnitudes should disrupt the balance
between of carbon sequestration and release in the peatland area.
The results highlight that DOC export magnitudes from the
permafrost catchment depend mainly on the discharge volume at the time
scale for the whole seasonal period. As noted above, our results prove that





DOC export concentrations vary considerably based on discharges of the
observed magnitude and frequencies of the three years while DOC
concentrations in the peatland area remain relatively stable across the
seasons. This finding is consistent with previous work suggesting that the
DOC export from permafrost organic soils is rainfall- and not
carbon-limited (Judd and Kling, 2002; Prokushkin et al. 2008; Olfeldt and
Roulet, 2012). Therefore, total rainfall levels are a robust predictor of
future pathways of DOC export from the catchment. It has been predicted
that precipitation in the study area will increase by 15.95% at most over
the next 50 years based on observational data on CN05 and based on the
outputs of 26 CMIP5 (Coupled Model Inter-comparison Project Phase 5)
models (Tao et al., 2016). Hence, the DOC export magnitude from the
permafrost is likely to increase following precipitation, which should
greatly enhance risks of losing an important active carbon pool in the
northern region of the Great Xing'an Mountains. As few data have been
generated of the southern margins of Eurasian permafrost to date, more
detailed investigations and data on the region are urgently needed to
evaluate future land carbon responses to climate change.

**5. Conclusions**
Eurasian permafrost serves as an important potential carbon pool for
the atmosphere and for linked aquatic and ocean ecosystems.





Investigations of DOC responses to permafrost peatland can be used to predict the ecological consequences of climatic change in these regions. Our study thoroughly explains the characteristics and determinants of DOC export from a peatland catchment along the southern margins of Eurasian permafrost. DOC magnitudes, sources, and chemical characteristics in stream discharge are greatly affected by runoff processes. Stable oxygen isotopes show that flood volumes and DOC exported in flood periods mainly derive from peat pore water stored in the upper organic layer of the soil profile prior to rainfall events. DOC concentrations are significantly related to steam discharge levels due to strong levels of hydrological connectivity between peatland areas and streams, thus rendering stream discharge (and flood volumes in particular) a strong indicator of DOC export magnitude. The three fluorescence indexes of HIX, FI and BIX show that DOC source and chemical characteristics change considerably with discharge processes. A deepening of active layer following permafrost degradation should increase levels of microbial-originated DOC content in baseflow discharge by elevating DOC contribution from the lower mineral soil layer. From our field data, the catchment exhibits strong potential for annual DOC export (4.87 g C/m$^2$), and DOC levels for the peatland landscape are estimated at 12.89 g C/m$^2$, representing 72.8% of the net ecosystem exchange (NEE). Given the potential for increases in





precipitation in the study region, DOC export levels are expected to
increase in the future, accelerating the loss of dissolved carbon pools
from East Asian permafrost.

**Acknowledgements**
The work was supported by National Key Research and Development
Program of China (2016YFA0602303), National Natural Science
Foundation of China (41571097), Key of Frontier Sciences, Chinese
Academy of Sciences (QYZDJ-SSW-DQC013), Research Program of
Northeast Institute of Geography and Agroecology, Chinese Academy of
Science (IGA-135-05).

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






**Table 1.** Correlation analysis of the three fluorescence indices with hydrological and
climatic factors.

|  |  | DOC | Q | Conductivity | Turbidity | $T_{air}$ | $T_{soil}$ |
|---|---|---|---|---|---|---|---|
| HIX | Pearson | 0.708** | 0.609* | 0.451** | −0.592** | 0.342 | 0.395* |
|  | Sig. (2-tailed) | 0.000 | 0.000 | 0.005 | 0.000 | 0.115 | 0.02 |
|  | n | 92 | 92 | 68 | 68 | 92 | 92 |
| FI | Pearson | −0.594** | −0.606** | −0.477** | 0.469** | 0.353 | 0.389 |
|  | Sig. (2-tailed) | 0.000 | 0.000 | 0.004 | 0.001 | 0.203 | 0.128 |
|  | n | 92 | 92 | 68 | 68 | 92 | 92 |
| BIX | Pearson | −0.64** | −0.707** | −0.488** | 0.322* | −0.027 | 0.384 |
|  | Sig. (2-tailed) | 0.001 | 0.000 | 0.001 | 0.012 | 0.823 | 0.129 |
|  | n | 92 | 92 | 68 | 68 | 92 | 92 |

DOC is dissolved organic carbon; Q is stream discharge; $T_{air}$ is the average air
temperature over the past three days; $T_{soil}$ is the average soil temperature of the active
layer; "**" denotes $p < 0.01$; "*" denotes $p < 0.05$





**Table 2.** Results of the correlation analysis of dissolved organic carbon (DOC) in the
soil pore water with soil factors

|  |  | SOM | TN | TP | Bulk density | Water content | $T_{soil}$ |
|---|---|---|---|---|---|---|---|
| **DOC** | Pearson | 0.733** | 0.602* | 0.341 | −0.671** | 0.337 | 0.492 |
|  | Sig. (2-tailed) | 0.000 | 0.02 | 0.187 | 0.005 | 0.144 | 0.07 |
|  | n | 18 | 18 | 18 | 18 | 18 | 18 |

SOM, TN and TP denote soil organic matter content, total nitrogen and phosphorus respectively;
$T_{soil}$ is the soil mean temperature at each depth; "**"denotes $p < 0.01$; "*" denotes $p < 0.05$.





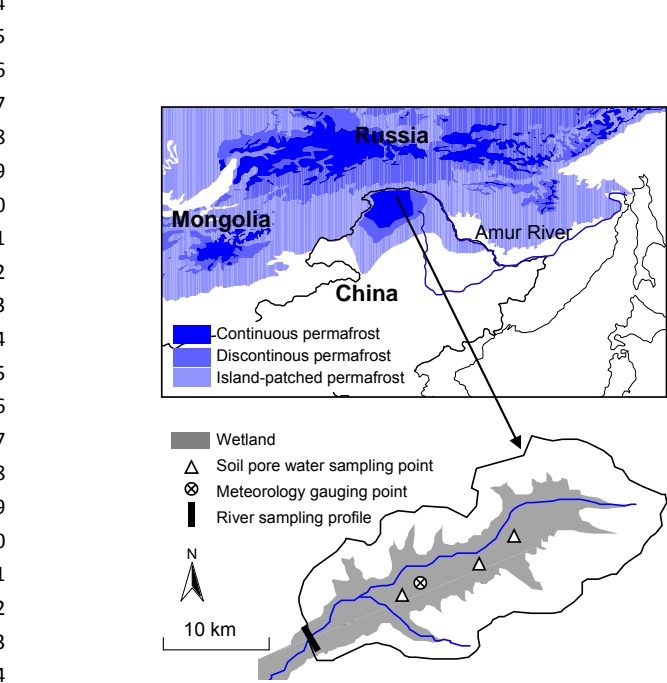

**Fig. 1** Geographic location of the study area





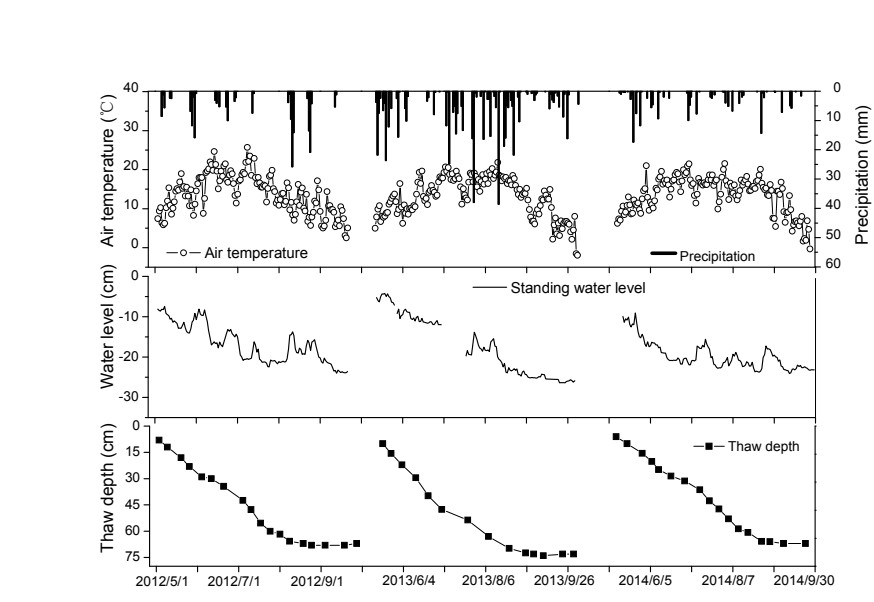

**Fig. 2** Dynamics of air temperature, precipitation, standing water levels, and thaw depth observed during the growing seasons of 2012 to 2014.




















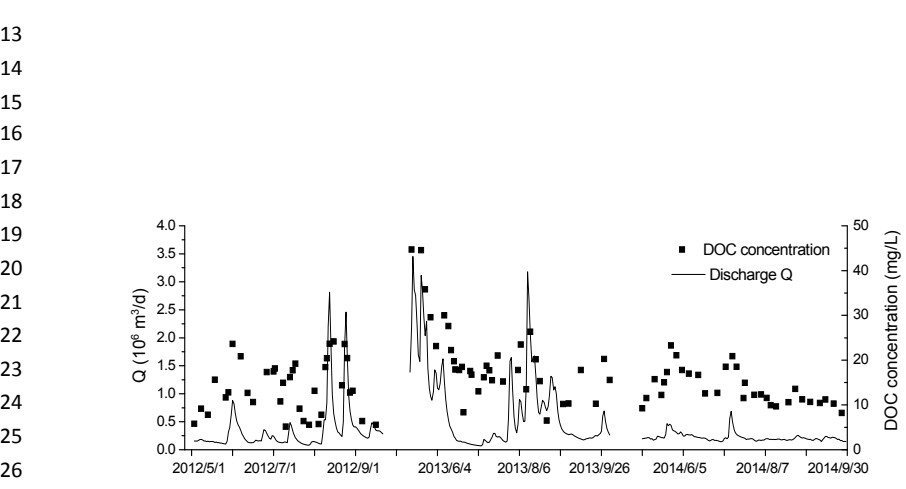

**Fig. 3** Dynamics of dissolved organic carbon (DOC) concentrations and discharge observed during the growing seasons of 2012 to 2014. The discharge (Q) unit used is $10^6$ m$^3$ d$^{-1}$.
















































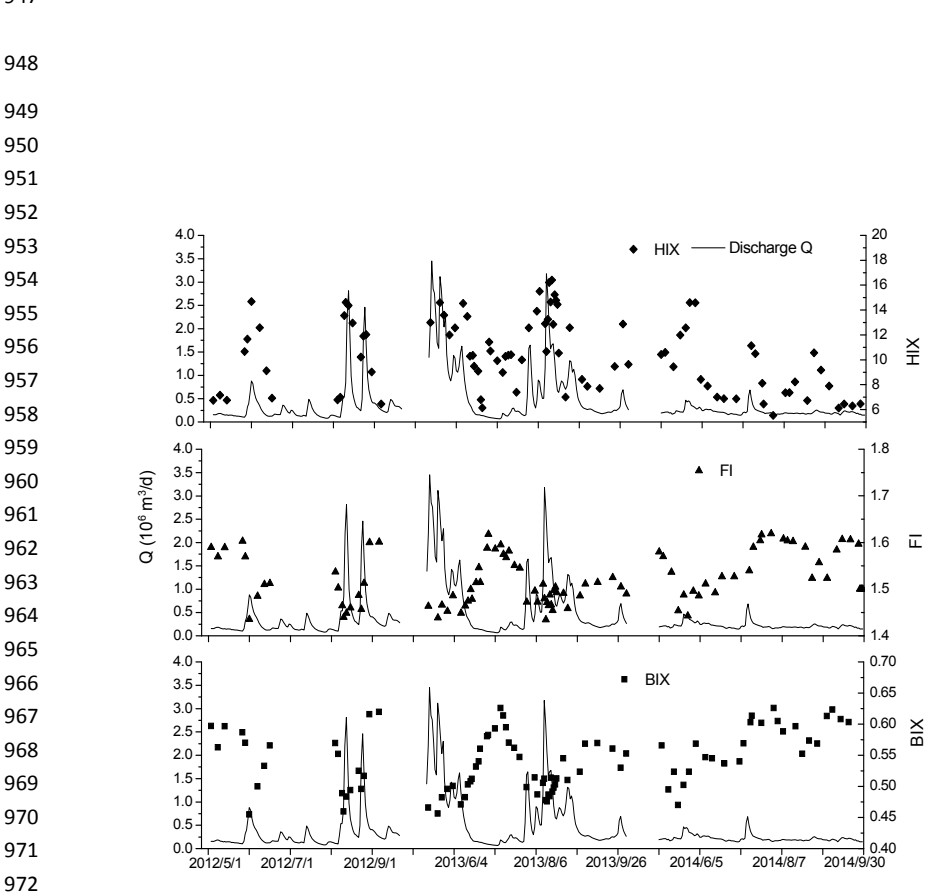

**Fig. 4** Relationships between discharge and the three spectral indexes during the

growing seasons.












































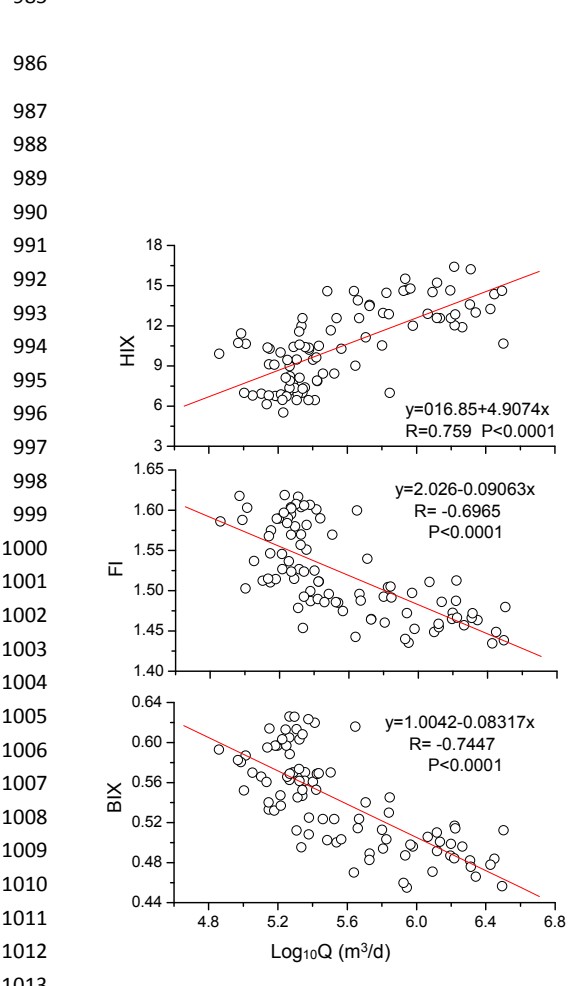

**Fig. 5** Relationships between discharge and the three indexes during the study period.















**Fig. 6** Dynamics of stable isotope oxygen values for rainfalls, discharge and soil pore

water in the catchment.















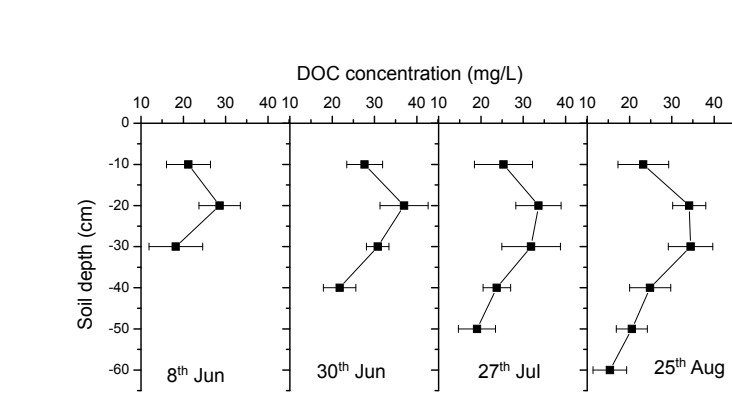

**Fig. 7** DOC concentrations in soil pore water along the soil profile for 2013.






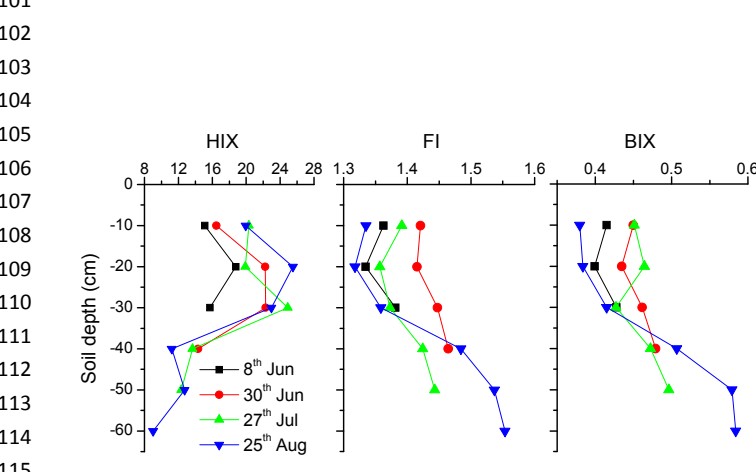

**Fig. 8** Vertical distribution of the three spectral indexes for soil pore water along the

soil profile for 2013.

