# Peer review of "Hydrological processes and permafrost regulate magnitude, source"

_Hydrology and Earth System Sciences, 2017_

## Referee Comment (RC1) · Anonymous Referee #1 · 14 Aug 2017

General comments: This paper summarizes a study of dissolved carbon dynamics in a peatland catchment in northeastern China underlain by continuous permafrost. Given that there are very few (if any) studies of C dynamics in this region, this work could be a worthwhile addition to the literature. Most of the studies to date have focused on far north latitudes, including Siberia, Alaska, and northern Scandinavia. This study can further the expansion of the world C database, and can only help aid in regional and global upscaling of C exports and yields. However, I have specific comments below for which clarification is advised. And while I applaud the authors' efforts to write in a nonnative language, I feel this article would benefit greatly from an English language edit. Once these items are addressed, I believe this article could be suitable for publication.

Specific comments: How did the authors calculate the DOC yield (4.87 g m-2 yr-1)? Were DOC concentrations and Q modeled in a program such as LOADEST to determine total export, or did the authors sum DOC*Q for all of the time points?

From the stated 4.87 g m-2 yr-1 yield, an extrapolation is made to a peat export of 12.89 g m-2 yr-1, but this is based on a faulty assumption that export is proportional to the C pool. This is not a valid assumption, as C is exported only if there is water flowing through the pool. While the authors state that this assumption leads to an overestimation, the peat export is extrapolated further to suggest that DOC export is 72.8% of NEE. I believe there is too much extrapolation based on that faulty assumption.

A major conclusion of the paper is that DOC concentration is highly correlated with discharge, and thus, DOC export is driven by discharge. While the relation was stated as highly significant, I would like to see a plot of DOC concentration vs. Q.

In the discussion, the authors first state that high infiltration rates prevent overland flow (lines 402-406). Later, the authors suggest that in spring when the upper soil is still frozen, overland flow through litter could be another DOC source. Please re-word to reconcile these statements. I am not sure I agree with the authors' definitions of "autochthonous" and "allochthonous." I think in most of the literature, autochthonous refers to DOC generated within the stream, while allochthonous refers to DOC generated outside of the stream. I'm guessing the authors have defined riparian peat as part of the stream, and hence by that definition they would be an autochthonous source. But by the definition in other studies, riparian peat would be a terrestrial source and hence allochthonous. And by this definition, the findings in this study would not be contrary to the view that the source of DOC from headwater catchments is allochthonous. So, I would just be careful of your definitions, or state them more clearly.

Sampling was conducted during the "growing season," May to September. There is no

discussion regarding the condition of the river during the rest of the year. Is it frozen the remainder of the year? Is there under ice groundwater flow? Section 2 refers to lost water quality data due to "excessively low temperatures in the stream." Did the sonde freeze in ice? How much data was lost?

The authors mention that peatland is distributed along both sides of the stream channel, and refer to Fig 1. The legend for Fig 1 refers to wetlands. Are the authors equating peatland and wetland? Also, as both upland and lowland are referred to in the text, I would like to know the elevation range in the catchment.

The authors say that permafrost blocks the input path of shallow groundwater from upland areas. Some mention of the difference in the depth to permafrost between upland and lowland areas would be good.

While the authors state their DOC yield (4.87 g m-2 yr-1) in the context of a range of values found in the literature (1-35 g m-2 yr-1), I would like to see more discussion of the difference in yields found between Eurasian studies and those of more northern latitudes. Is it typical for Eurasian catchments to be on the low end of the world average? And I suspect that there are more recent references with DOC yields than 2010.

I would like to see error bars on Fig. 8, so that it is easier to tell if there is any real seasonal difference in the spectral profiles.

Specific corrections: Check all units for superscripts and SI units in both text and figures. In some instances I see mg/L, in others mg L-1, etc.

Be consistent with "indexes" and "indices."

How frequently were water level and flow velocity measured?

How frequently were temperature, conductivity, and turbidity measured?

For 18O analysis, please include a method reference.

Samples were collected from the "surface, middle, and bottom layers" of the profile.

What are the depths of these layers?

For DOC analysis, please include a method reference.

DOC analysis was done "as soon as possible." Within what time were they analyzed? Were they stored chilled before analysis?

The results refer to total soil organic matter, total nitrogen content, and soil bulk density, but there is no mention of these analyses in the methods section.

Line 67-68: I believe the Striegl reference is 2005, not 2007

Line 117: is there a reference for long term temperature rise in this area?

Line 171: How do you define a "flood period?"
* * *

---

## Referee Comment (RC2) · D. Olefeldt (Referee) · 28 Aug 2017

Review of "Hydrological processes and permafrost regulate magnitude, source and chemical characteristics of dissolved organic carbon export in a peatland catchment of northeastern China", by Guo et al.

**Main points:**

The manuscript presents results from a catchment study on DOC export, carried out in a permafrost affected catchment in northern China. The authors monitored magnitude, timing, and chemical composition of DOC over three years. The topic is of current interest, as understanding impacts of climate change on DOC export in northern catchments can have potential important implications for downstream carbon cycling and landscape carbon cycling. Furthermore, as far as I know this is one of the first studies on this topic from eastern Asia, as most current research has been carried out in Scandinavia or North America, and thus is of particular interest. Three years of data, from highly different hydrological years, also increase the interest of the data-set.

My main comments on the manuscript are A) that I find that there is substantial room for improvement on the writing style, B) that the data analysis can be improved, particularly with regards to including uncertainty estimates and exploring influences of different seasons/years, and C) that I find that the authors are not sufficiently highlighting what I find to be the more interesting results.

I find that the authors use vague terms throughout the manuscript. For example, the authors talk about "runoff processes" and "hydrological processes" as the main controls on DOC export, but never clearly define what processes in particular they are referring to. For example, in several places I think it could be better to talk about "shifting flow-paths", rather than use the more general term "hydrological processes".

The writing style is generally indirect, unnecessarily wordy, and with main points found at the end of paragraphs or sections. Often I found this made it hard to follow the logic of a paragraph until I had arrived at the main point. I recommend writing using an active voice, and to adopt a style where the main point of a section or a paragraph is indicated early rather than at the end (i.e. point first paragraphs). I also found that many sentences can be re-written much shorter without losing the main message.

I find that the data analysis and reporting could be improved. Throughout the manuscript I find that the authors are reporting data with too high precision, using too many significant digits. At the same time, none of the estimates e.g. of DOC export, have uncertainty bounds. The methodology for estimating DOC export is never explained. There are many methods for estimating loads, and many of them will also yield uncertainty bounds. I also think there is a missed opportunity in the data analysis, to run an analysis of covariance (ANCOVA) to see if relationships between discharge and the several DOC characteristics (concentration and fluorescence indices) are statistically different for different seasons or years. If the relationship between discharge and DOC characteristics is dependent on e.g. active layer

depth (season), that would be really interesting. I think there are some general qualitative comments along these lines, and it would be very good to show this explicitly.

I find that the authors are emphasizing results/conclusions that are already well established, while perhaps missing an opportunity to highlight some of the findings that I consider more novel. The novel finding to me include the fact that these wetlands do not seem to behave as boreal peatlands in Scandinavia or North America, and thus that we could expect a different response to permafrost thaw at the study catchment. In peatland catchments in boreal Scandinavia or North America, we usually observe that DOC concentrations decrease during periods of higher discharge – a dilution effect. At the studied catchment, the authors report the opposite pattern. As such the studied peatland catchment act much more as an upland catchment where shifts in DOC concentrations and its chemical characteristics is controlled by riparian hydrology, and the shifts in flowpaths from organic surface soils during high flow periods and deeper mineral soil flow-paths during low flow periods. This seems to follow from the shallow peat layer at the studied catchment – and the study nicely shows that there are important shifts in DOC characteristics at the interface between the organic and mineral soil layers. With a deeper active layer, which develops earlier in the season, this would mean that we can expect an increased importance of deeper flow-paths through mineral soils – with associated shifts in DOC export. This to me represents a novel finding, and it contrasts very nicely with results from other regions.

 **Specific points:**

L20. Specify whether you mean organic matter degradation or permafrost degradation.

L26. Export at ~4.5 g C m-2 yr-1 is hardly "strong potential" – it is relatively moderate catchment DOC exports.

L49. This sentence:  "However, uncertainties remain regarding to main driving factors involved and the fate of DOC due to complex interactions between hydrological and thermal dynamics and bio-chemical drivers." – what is meant by thermal dynamics and bio-chemical drivers in this context? I think this is a good example of where the authors are vague, and where more specific examples would help convey their message better.

L84. Another example: "However, uncertainties remain in predicting DOC export processes based on changing hydrological processes." – This is vague since I don't know what you refer to when you say "hydrological processes".

L90. DOC export is not only a function of runoff (if that is what is meant by with hydrological regime"). Catchment land-cover, particularly peatland land cover, has repeatedly been shown to influence catchment DOC export.

L97. Replace "can stably" with "preferentially"

L96-101. Long sentence – split into at least 2 sentences.

L143. What are the permafrost conditions in the catchments? Continuous or confined to the peatlands?

L292. How are these values calculated? Arithmetic means based on all sampling occasions? Or is there an adjustment based on the discharge at the time of sampling?

L293. Always write out units as mg $L^{-1}$, not mg/L.

L296. Use a SI unit to report mass C export, not *t*.

L297. How was DOC export calculated? There are several methods for estimating DOC export, and with many of them you can also estimate your uncertainty. Right now you are stating the DOC export with very high precision and no error estimates. Simple methods are outlined in Walling and Webb (1985 Marine Pollution Bulletin).

L297. Remove "Statistically speaking" as it is colloquial.

L297-302. Here it would be good if you could say for how long these events lasted. I.e. 85% of the DOC export occurred during X% of the time of the monitoring program.

L313. These correlations with discharge – were there distinct correlations during each year or during each season? You could carry out an ANCOVA to see whether slopes or intercepts of relationships differed for different periods.

L389-392. I don't see how the first and second parts of this sentence are connected.

L416-425. I find that this discussion on runoff sources and relationship between discharge and DOC concentrations needs to be explicit about what water sources are considered to dominate stream flow during low-flow periods. In boreal catchments

Also, this positive relationship between DOC concentration and discharge is opposite what is generally found in boreal peatland catchments in Scandinavia and North America. What is different?

L452. I do not agree that peatland-derived DOC should be considered "autochtonuous". Usually that term implies DOC derived from aquatic primary production, and is contrasted to "allochtonous" DOC derived from the terrestrial surroundings.

L511-526. This seems to me like one of the most interesting findings of the study, and could be elaborated on more.

L538. The word "synthetically" is not appropriate – I believe you mean something like "through synthesis".

L555. I don't agree that DOC export among ecosystems within a catchment need to be proportional to the soil organic C stock of each ecosystem. There is no data to support this assumption.

L563. The term "verified NEE" should not be used. Perhaps you can use the term Net Ecosystem Carbon Balance (NECB), but then you also need to take into account methane-C emissions.

L579. This conclusion does not take into account shifts in water quality, including DOC concentrations, due to deepened flow-paths following active layer deepening through the mineral soil under the 30-40 cm organic soil.

L581. Stating too high precision on projected increases in precipitation. Also, what are the projected changes to evapotranspiration? If ET increases more than precipitation, then we would expected reduced runoff of course.

L616. The definition of NEE only includes the land-atmosphere exchange of $CO_2$ – you have further included DOC export in this NEE estimate that you compare DOC export to, which is not correct.

Fig7 and 8. I would recommend that you indicate at what depth the soil type changes from peat to mineral.

---

## Referee Comment (RC3) · Anonymous Referee #3 · 9 Sep 2017

Guo et al utilize a data set of stream discharge, carbon concentration and quality, and various other ancillary pieces of information from near stream catchment peatlands to identify relationships between discharge and DOC in a catchment underlain by continuous permafrost in northern China. The discharge and DOC data set is robust, indicating a direct correlation between these two variables. They argue that the main source of the DOC is peatland soils and that the forested hillslopes do not contribute to the signal. Three measures of carbon quality are used to support their claim and to show the similarity between stream and soil chemistry. No chemical or physical data is

presented for the forested parts of the catchment.

DOC is difficult to measure during storms, and the authors have done a great job of capturing the rise and recession of several storms - this in itself is a great contribution. Generally, there is more that could be done with the available data to support the authors' claims and investigate the relationship between discharge and DOC. For instance, plotting discharge vs. DOC could help understand the relationship between these two and aid in considering a wetter future. Breaking the data set up by years could also be useful, given the large differences in precipitation between the years. There is lots of speculation about the source of the carbon and the flowpaths between the catchment and the stream. I think this speculation significantly detracts from the paper. The authors' analysis relies heavily on several references from other systems, which may or may not be representative of conditions in their system. The authors do a fair job of addressing their second research question regarding the relationship between runoff processes and DOC, but do not – and probably are not capable of rigorously answering their third question regarding the effect of permafrost degradation and climate change given the data set.

Major Comments:

Connectivity vs variable source areas – what is the ultimate cause of the observed trends between discharge and DOC and the fluorescence indices? – the authors argue that the thaw depth controls DOC export concentration and quality, and make some assumptions about contributing areas and catchment connectivity, but with minimal support beyond a few references from other well-known catchments studies. The authors do not have any data from the forested hillslopes, which is an important endmember necessary to substantiate their claims. Especially the paragraph from 428 – 450 appears highly speculative

Generally, I found the paper to be very light on analysis - a major finding is that DOC is positively correlated to discharge. It would be nice to see this relationship plotted.

Is it linear? Non-linear? Showing this would add further support to the author's claim that this system is transport-limited and that increased rainfall will lead to increased C export.

Consistency in terms – at line 316 – "hydrological DOC, Q, conductivity, and turbidity', earlier at line 288, "discharge turbidity" and "discharge conductivity"

Unclear interpretation of FI index – FI varies in the soils from 1.3 to 1.55 and in the stream from 1.43 to 1.62. The authors assume that the range indicates "...both terrestrial and microbial sources" (line 320), and cite Cory's 2010 paper which focuses on correcting fluorescence spectra from different instruments. This is not an appropriate reference to support the authors' interpretation. McKnight's 2001 publication in which microbial and terrestrial end members is defined would be a better choice, but still it would be useful to present more rationale for the authors' interpretation, and to address alternate hypotheses to explain the differences – like the influence of distal water sources.

Stable isotopes – the authors argue that stable isotopes indicate that peat porewaters, rather than direct rainfall or mineral soils are the source of runoff. But they have not measured isotopes from the mineral soils beneath the peats, or from the more distal parts of the catchment (ie. the hillslopes), and thus the argument is weak. Furthermore, it is unclear from Figure 6 whether they've collected enough samples to see changes to the isotopic composition for individual storms, which limits the potential inference.

Unclear how total DOC export magnitudes were estimated – there are no methods regarding the calculations used.

Minor Comments:

56 – missing some major references regarding the effects of permafrost thaw on hydrology, for instance Hinzman et al., 2005, Jorgenson et al., 2006. ...

101 – 103: How do you define 'satisfactory'? Many studies have focused on the fate of

permafrost carbon – Drake et al., 2015 is a good example and much of the BDOC loss methods by Wickland and others. Spencer et al., 2015 provides a nice conceptualization of the fate of permafrost carbon.

128 – This is very broad question involving multiple disciplines and I'm not convinced that your data set is nearly enough to address this. I would suggest removing this question all together.

161: What is 'yang'?

177: How soon is '…as soon as possible.'? Hours? Days? Weeks? This matters especially when you're talking about DOC. Were samples stored in a cool, dark place before analysis?

183 and 190 – sensors are not consistently or properly referenced (ie. Campbell, USA and YSI6600, USA are both incomplete)

203: Missing a verb – maybe 'collected'?

271: This sentence is confusing. Do you mean that there was no standing water in the peat?

288: I think you can remove the word 'discharge' and just say 'turbidity'. Similarly, 'electrical conductivity' is clearer than 'discharge conductivity'.

319: Is it reasonable to assume that the FI range will be similar in your system to those studied by Cory?

347: If they were not statistically different, wouldn't the p value by larger than the 0.01 test threshold?

389: This sentence is not clear, and not totally true

397: This statement may be generally true, but not always. Organic soil macropores may not exist everywhere. Do they exist in the Fukuqi catchment? The high hydraulic

conductivity and porosity of shallow soils relative to deeper soils also plays an important role.

407: This may be evidence, I don't think that it's necessarily proof.

416: What do you mean by 'fundamental condition'?

418 – 420: I do not believe that subsurface flow "guarantees" that water closest to the stream will always reach the stream first. When subsurface conditions are homogeneous, this may be true, however soil pipes in organic soils (Carey and Woo 2001) and mineral soils (Koch et al., 2013), tussocks (Quinton et al., 2000), and hetereogeneity in subsurface soils (Koch et al. 2017; Laine-Kaulio 2014 and 2015 ) may complicate this and lead to preferential areas of flow, allowing some areas further from the stream to contribute faster and more than areas near the stream.

420: I don't quite follow this sentence. Maybe break it down into a few sentences? Also, the positive correlation between Q and DOC is likely a result of more dynamics than simply the proximity of organic-rich soils, it also implies that the source is large, and that the presence in the stream is transport-limited. This point seems important to the story you're telling. . .

428: Spence and Woo 2006 and Spence and Phillips 2015 both support this point and provide useful precedent.

431: "Geomorphic landscape structures" is kind of vague.

436: This sentence is unclear – if the peatland is highly conductive, shouldn't it facilitate movement of water from the hills?

441: I don't think you can assume the values of the hillslopes. I would not expect them to look more like rain than the peatland soil porewaters.

484: Where is the evidence that these don't generally change with DOC concentration?

451: I believe that your data very much supports an allochthonous DOC source –

autochthonous DOC would result from in-stream processes like the degradation of photosynthetic cells. Variations in contributing area also likely play an important role (see spence).

505 – 510: Koch et al. 2014 found that stream chemistry changed much earlier, around mid-June concurrent with the beginning of thawing of the mineral soils. Based on your depth to ice measurements and statement that the organic-mineral boundary is near 30 – 40 cm, it seems like you should also start seeing this response somewhere in June. Autumn sounds too late – by this time you've reached maximum thaw and in fact may be beginning to freeze again.

536: I don't understand the logic here. . ..how can you suggest that HIX values are not sensitive to soil active layer depths when you show substantial variations in HIX with soil depth (Figure 8)?

555: There are two assumptions here and I'm not sure if either is reasonable: 1. Is it reasonable to assume that export is proportional to concentration for both forest and peatland systems? Forest and peatland carbon is fairly different, and I imagine could have differing levels of leachability and solubility, and thus transport potential. And I don't believe that you've discussed forests at all before this point. 2. This seems to ignore your previous claims that only the peatland contributes to the stream DOC pool.

587 – But at the same time temperatures are likely to warm, impacting overall carbon stocks and DOC production. So there are lots of variables that will likely affect the active carbon pool.

Figure 1 needs lat/longs

Fig 2 – Date format is difficult to read and in strange increments. What does 'standing water level' mean? Is this level and the thaw depth from one point? How representative is this point? Is this point shown in Figure 1?

Fig 6 – It would be nice to also have discharge on this plot to see how stream water

isotopes relate to discharge.

Fig 7 – Probably don't need negatives on the y axis – What would a negative soil depth mean? Why not set up this plot like those in Figure 8? It would make it easier to compare the seasonal trends.

References mentioned in this review Drake, T. W., K. P. Wickland, R. G. Spencer, D. M. McKnight and R. G. Striegl (2015). "Ancient low–molecular-weight organic acids in permafrost fuel rapid carbon dioxide production upon thaw." Proceedings of the National Academy of Sciences 112(45): 13946-13951.

Hinzman, L. D., N. D. Bettez, W. R. Bolton, F. S. Chapin, M. B. Dyurgerov, C. L. Fastie, B. Griffith, R. D. Hollister, A. Hope, H. P. Huntington, A. M. Jensen, G. J. Jia, T. Jorgenson, D. L. Kane, D. R. Klein, G. Kofinas, A. H. Lynch, A. H. Lloyd, A. D. McGuire, F. E. Nelson, W. C. Oechel, T. E. Osterkamp, C. H. Racine, V. E. Romanovsky, R. S. Stone, D. A. Stow, M. Sturm, C. E. Tweedie, G. L. Vourlitis, M. D. Walker, D. A. Walker, P. J. Webber, J. M. Welker, K. S. Winker and K. Yoshikawa (2005). "Evidence and implications of recent climate change in Northern Alaska and other Arctic regions." Climatic Change 72(3): 251-298.

Jorgenson, M. T., Y. L. Shur and E. R. Pullman (2006). "Abrupt increase in permafrost degradation in Arctic Alaska." Geophysical Research Letters 33(2).

Koch, J. C., S. A. Ewing, R. Striegl and D. M. McKnight (2013). "Rapid runoff via shallow throughflow and deeper preferential flow in a boreal catchment underlain by frozen silt (Alaska, USA)." Hydrogeology Journal 21(1): 93-106.

Koch, J. C., C. P. Kikuchi, K. P. Wickland and P. F. Schuster (2014). "Runoff sources and flow paths in a partially burned, upland boreal catchment underlain by permafrost." Water Resources Research 50(10): 8141-8158.

Laine‐Kaulio, H., S. Backnäs, T. Karvonen, H. Koivusalo and J. J. McDonnell (2014). "Lateral subsurface stormflow and solute transport in a forested hillslope: A

combined measurement and modeling approach." Water Resources Research 50(10): 8159-8178.

Laine-Kaulio, H., S. Backnäs, H. Koivusalo and A. Laurén (2015). "Dye tracer visualization of flow patterns and pathways in glacial sandy till at a boreal forest hillslope." Geoderma 259: 23-34.

Quinton, W. L., D. M. Gray and P. Marsh (2000). "Subsurface drainage from hummock-covered hillslopes in the arctic tundra." Journal of Hydrology 237(1-2): 113-125.

Spence, C. and M.-k. Woo (2006). "Hydrology of subarctic Canadian Shield: heterogeneous headwater basins." Journal of Hydrology 317(1): 138-154.

Spencer, R. G., P. J. Mann, T. Dittmar, T. I. Eglinton, C. McIntyre, R. M. Holmes, N. Zimov and A. Stubbins (2015). "Detecting the signature of permafrost thaw in Arctic rivers." Geophysical Research Letters 42(8): 2830-2835.

---

## Author Comment (AC1) · 31 Oct 2017

Comment 1: How did the authors calculate the DOC yield (4.87 g m-2 yr-1)? Were DOC concentrations and Q modeled in a program such as LOADEST to determine total export, or did the authors sum DOC*Q for all of the time points? Response: Thanks for the comment! LOADEST program is a good tool to estimate total DOC yield. In the revised paper, the DOC yield was re-calculated by the program LOADEST with the web-based calculation program (https://engineering.purdue.edu/mapsever/ldc/LOADEST, version 2012). The new DOC yield estimated by the program is 4.7 g m-2 yr-1, which

is very close to the previous result. The previous result, 4.87 g m-2 yr-1, is obtained by multiplying seasonal mean DOC concentration by total discharge. The new result will used in the revised paper (Lines 219-232; Line 302).

Comment 2. From the stated 4.87 g m-2 yr-1 yield, an extrapolation is made to a peat export of 12.89 g m-2 yr-1, but this is based on a faulty assumption that export is proportional to the C pool. This is not a valid assumption, as C is exported only if there is waterflowing through the pool. While the authors state that this assumption leads to an overestimation, the peat export is extrapolated further to suggest that DOC export is 72.8% of NEE. I believe there is too much extrapolation based on that faulty assumption. Response: Thanks for the comment! Indeed, the data of 72.8% is indeed overestimated. The content about the data is deleted in the revised paper, and the discussions about the importance of DOC yield for the peat carbon pool is re-organized (Lines 427-435).

Comment 3: A major conclusion of the paper is that DOC concentration is highly correlated with discharge, and thus, DOC export is driven by discharge. While the relation was stated as highly significant, I would like to see a plot of DOC concentration vs. Q. Response: Thanks for the comment! A new figure (Figure 4) showing the relationship was added in the revise paper. In the new figure, there exhibits a significantly positive relationship between DOC concentration and log10 (Q). (Page 45) Comment 4. In the discussion, the authors first state that high infiltration rates prevent overland flow (lines 402-406). Later, the authors suggest that in spring when the upper soil is still frozen, overland flow through litter could be another DOC source. Please re-word to reconcile these statements. Response: Thanks for the comment! This part was deleted in the revised paper because it is only a conjecture without support of field data. Meanwhile, the discussion section was largely re-written, and the part in the new paper is not necessary.

Comment 5. I am not sure I agree with the authors' definitions of "autochthonous" and "allochthonous." I think in most of the literature, autochthonous refers to DOC generated within the stream, while allochthonous refers to DOC generated outside of the stream. I'm guessing the authors have defined riparian peat as part of the stream, and hence by that definition they would be an autochthonous source. But by the definition in other studies, riparian peat would be a terrestrial source and hence allochthonous. And by this definition, the findings in this study would not be contrary to the view that the source of DOC from headwater catchments is allochthonous. So, I would just be careful of your definitions, or state them more clearly. Response: Thanks for the comment! In the revised paper, the expressions of "autochthonous" and "allochthonous" was deleted to avoid confusion. The discussion about DOC origin is re-organized in the revised paper (Lines 467-480).

Comment 6. Sampling was conducted during the "growing season," May to September. There is no discussion regarding the condition of the river during the rest of the year. Is it frozen the remainder of the year? Is there under ice groundwater flow? Section 2 refers to lost water quality data due to "excessively low temperatures in the stream." Did the sonde freeze in ice? How much data was lost? Response: Thanks for the comments. The information about the rest of the year was replenished in the section "2.1. Study area". Low temperature had led to the power loss of the buttery in the sonde. There was no ice forming during the growing seasons. Totally, about one fifth of the water quality data was lost mostly in early May in 2012 and 2013. (Lines 153-154 in the revise paper).

Comment 7: The authors mention that peatland is distributed along both sides of the stream channel, and refer to Fig 1. The legend for Fig 1 refers to wetlands. Are the authors equating peatland and wetland? Also, as both upland and lowland are referred to in the text, I would like to know the elevation range in the catchment. Response: In the revised paper, the legend has changed into "Peatland" in figure 1. The elevation range of upland and lowland was added in the section "2.1. Study area". The description about the landform of the whole catchment was replenished in the section in lines119-122 in the revise paper.

Comment 8: The authors say that permafrost blocks the input path of shallow groundwater from upland areas. Some mention of the difference in the depth to permafrost between upland and lowland areas would be good. Response: The maximum thaw depth of the upland ranges from 80 to 100cm, which is slightly deeper than that in the peatland. This information about the thawing depth of upland was added in the revised paper in lines 138-139. However, the content about hydrological connectivity between mountain and peatland river is deleted due to lacking of the support from field data according to the comment from another reviewer.

Comment 9: While the authors state their DOC yield (4.87 g m-2 yr-1) in the context of a range of values found in the literature (1-35 g m-2 yr-1), I would like to see more discussion of the difference in yields found between Eurasian studies and those of more northern latitudes. Is it typical for Eurasian catchments to be on the low end of the world average? And I suspect that there are more recent references with DOC yields than 2010. Response: The references about the DOC yield from Eurasian and other northern sites were re-concluded in the revised paper. New references after 2010 were included and typical examples from boreal land was added in the revised paper in lines 412-420.

Comment 10: I would like to see error bars on Fig. 8, so that it is easier to tell if there is any real seasonal difference in the spectral profiles. Response: The error bars was added in Fig. 9 (Fig. 8 in original paper) in the revised paper (Page 50).

Comment 11: Check all units for superscripts and SI units in both text and figures. In some instances I see mg/L, in others mg L-1, etc. Be consistent with "indexes" and "indices." Response: The errors in the units and other language descriptions have been all modified in the revised paper.

Comment 12: How frequently were water level and flow velocity measured? How frequently were temperature, conductivity, and turbidity measured? Response: All the data are set to be measured once every six hours. This information has added to the
revised paper (Lines185).

Comment 13: Samples were collected from the "surface, middle, and bottom layers" What are the depths of these layers? Response: Thanks for the comment! As the water level in the gauging profile of the stream fluctuated with time, so no exact depths was recorded when collecting water samples. There must be some mistakes in my description about water sampling, and the sample process was re-written in the revised paper in lines148-150.

Comment 14: For DOC analysis, please include a method reference. Response: Thanks for the comment. A reference on the DOC measurement with the same DOC analyser was added in the revised paper (Line 153).

Comment 15: DOC analysis was done "as soon as possible." Within what time were they analyzed? Were they stored chilled before analysis? Response: Thanks for the comment! The information on how to store the samples was added in the revised paper in lines151-152.

Comment 16: The results refer to total soil organic matter, total nitrogen content, and soil bulk density, but there is no mention of these analyses in the methods section. Response: The related content was deleted in the revise paper, because the analysis about the relationship between soil DOC and soil features is no use to explain the DOC dynamics in the stream.

Comment 17: Line 67-68: I believe the Striegl reference is 2005, not 2007 Response: Thanks for the comment! The information was modified in the revised paper in line 61.

Comment 18: Line 117: is there a reference for long term temperature rise in this area? Response: The sentence was revised and a reference was added in the line 97.

Comment 19: Line 171: How do you define a "flood period?" Response: After rainfalls, the water level in the stream profile would rise and go down. A flood period just means a flood event. The "flood period" maybe not clear, it was replaced by "flood events".

Please also note the supplement to this comment:
https://www.hydrol-earth-syst-sci-discuss.net/hess-2017-412/hess-2017-412-AC1-supplement.zip

---

## Author Comment (AC2) · 31 Oct 2017

The comments from reviewer 2:

Comment 1. I find that there is substantial room for improvement on the writing style. Response: Thanks for the comment! The paper was largely revised according to the comments, and it had been re-edited carefully to make the writing style more clear.

Comment 2. The data analysis can be improved, particularly with regards to including uncertainty estimates and exploring influences of different seasons/years. Response:

[Figure]

Thanks for the comment! DOC loads was re-calculated and the data, as well as the uncertainty estimates, in different years and seasons was listed in table 2 (Page 39).

Comment 3. I find that the authors are not sufficiently highlighting what I find to be the more interesting results. Response: Our results have compared with other research in North America in the revised paper, and the reasons leading to the different results have been discussed extensively. (In Section 4, Lines 370-444 in the revised paper).

Comment 4. I find that the authors use vague terms throughout the manuscript. For example, the authors talk about "runoff processes" and "hydrological processes" as the main controls on DOC export, but never clearly define what processes in particular they are referring to. For example, in several places I think it could be better to talk about "shifting flow-paths", rather than use the more general term "hydrological processes". Response: Thanks for the comment! The vague terms were replaced by exact expressions such as "shifting flow-paths" in the revised paper.

Comment 5. The writing style is generally indirect, unnecessarily wordy, and with main points found at the end of paragraphs or sections. Often I found this made it hard to follow the logic of a paragraph until I had arrived at the main point. I recommend writing using an active voice, and to adopt a style where the main point of a section or a paragraph is indicated early rather than at the end (i.e. point first paragraphs). I also found that many sentences can be re-written much shorter without losing the main message. Response: Thanks for the comment! I re-written some paragraph according the comments especially the discussion sections. The main points of the sections were presented in the beginning of the paragraph.

Comment 6. I find that the data analysis and reporting could be improved. Throughout the manuscript I find that the authors are reporting data with too high precision, using too many significant digits. At the same time, none of the estimates e.g. of DOC export, have uncertainty bounds. The methodology for estimating DOC export is never explained. There are many methods for estimating loads, and many of them

will also yield uncertainty bounds. I also think there is a missed opportunity in the data analysis, to run an analysis of covariance (ANCOVA) to see if relationships between discharge and the several DOC characteristics (concentration and fluorescence indices) are statistically different for different seasons or years. If the relationship between discharge and DOC characteristics is dependent on e.g. active layer depth (season), that would be really interesting. I think there are some general qualitative comments along these lines, and it would be very good to show this explicitly. Response: Thanks for the comment! First, the DOC loads were re-estimated by a new method according to the suggestion from the first reviewer. In the revised paper, the DOC load was re-calculated by the program LOADEST with the web-based calculation program (https://engineering.purdue.edu/mapsever/ldc/LOADEST, version 2012). The new result will used in the revised paper (In lines 219-232; Line 302). Second, the relationship between discharge and DOC concentrations in each year was plotted in Figure 4 (Page 45 in the revised paper). Third, the analysis about the relationships between discharge and the three indexes was supplied in Table 1 and Table 4 (Page 38 and 41). The result shows that the discharge is the sole controlling factor on DOC inter-annual variation, but not the sole factor for the mean DOC concentration. The inter-seasons analysis was not conducted for lacking enough data for each month.

Comment 7. I find that the authors are emphasizing results/conclusions that are already well established, while perhaps missing an opportunity to highlight some of the findings that I consider more novel. The novel finding to me include the fact that these wetlands do not seem to behave as boreal peatlands in Scandinavia or North America, and thus that we could expect a different response to permafrost thaw at the study catchment. In peatland catchments in boreal Scandinavia or North America, we usually observe that DOC concentrations decrease during periods of higher discharge – a dilution effect. At the studied catchment, the authors report the opposite pattern. As such the studied peatland catchment act much more as an upland catchment where shifts in DOC concentrations and its chemical characteristics is controlled by riparian hydrology, and the shifts in flowpaths from organic surface soils during high flow peri-

ods and deeper mineral soil flow-paths during low flow periods. This seems to follow from the shallow peat layer at the studied catchment – and the study nicely shows that there are important shifts in DOC characteristics at the interface between the organic and mineral soil layers. With a deeper active layer, which develops earlier in the season, this would mean that we can expect an increased importance of deeper flow-paths through mineral soils – with associated shifts in DOC export. This to me represents a novel finding, and it contrasts very nicely with results from other regions.

Response: Thanks for the comment! The DOC concentrations, loads and the relationship with discharge was extensively compared with other studies in boreal area including North America in the revised paper (In lines 412-420 in the revised paper). The influence of active layer deepening on DOC chemical characteristics was also carried out in the discussion section in the revised paper (Lines 510-537).

L20. Specify whether you mean organic matter degradation or permafrost degradation. Response: "permafrost degradation" was added in the revised paper in lines 20.

L26. Export at ∼4.5 g C m-2 yr-1 is hardly "strong potential" – it is relatively moderate catchment DOC exports. Response: The DOC loads were re-estimated to be 4.7 g C m-2 yr-1 which is in the lower range of boreal DOC yields (Lines 412-414 in the revised paper).

L49. This sentence: "However, uncertainties remain regarding to main driving factors involved and the fate of DOC due to complex interactions between hydrological and thermal dynamics and bio-chemical drivers." – what is meant by thermal dynamics and bio-chemical drivers in this context? I think this is a good example of where the authors are vague, and where more specific examples would help convey their message better. Response: This part was re-written in the revised paper. (In lines 48-50). The paper has been largely re-edited to avoid vague expressions.

L84. Another example: "However, uncertainties remain in predicting DOC export processes based on changing hydrological processes." – This is vague since I don't know

what you refer to when you say "hydrological processes". Response: This sentence was deleted in the revised paper.

L90. DOC export is not only a function of runoff (if that is what is meant by with hydrological regime"). Catchment land-cover, particularly peatland land cover, has repeatedly been shown to influence catchment DOC export. Response: This sentence was deleted in the revised paper, and some conclusions about the influence of flow path on DOC export were supplied in the introduction section (In lines 52-68 in the revised paper).

L97. Replace "can stably" with "preferentially" Response: This sentence was replaced in the revised paper in line74.

L96-101. Long sentence – split into at least 2 sentences. Response: The sentence was split into 2 sentences (Line 74-78).

L143. What are the permafrost conditions in the catchments? Continuous or confined to the peatlands? Response: The permafrost in the catchment is continuous. This information is added in the revise paper in lines 116, 138-139.

L292. How are these values calculated? Arithmetic means based on all sampling occasions? Or is there an adjustment based on the discharge at the time of sampling? Response: The DOC loads were re-estimated by a new method according to the suggestion from the first reviewer. In the revised paper, the DOC load was re-calculated by the program LOADEST with the web-based calculation program (https://engineering.purdue.edu/mapsever/ldc/LOADEST, version 2012).

L293. Always write out units as mg L-1, not mg/L. Response: Thanks for the comment! The errors were all modified in the revised paper.

L296. Use a SI unit to report mass C export, not t. Response: Thanks for the comment! The unit was altered to the unit Kg.

L297. How was DOC export calculated? There are several methods for estimating

[Figure]

DOC export, and with many of them you can also estimate your uncertainty. Right now you are stating the DOC export with very high precision and no error estimates. Simple methods are outlined in Walling and Webb (1985 Marine Pollution Bulletin). Response: The DOC loads were re-estimated by a new method according to the suggestion from the first reviewer. The DOC load was re-calculated by the program LOADEST with the web-based calculation program (Lines 219-232 in the revised paper). L297. Remove "Statistically speaking" as it is colloquial. Response: The two words were deleted in the revised paper.

L297-302. Here it would be good if you could say for how long these events lasted. I.e. 85% of the DOC export occurred during X% of the time of the monitoring program. Response: Thank for the comment. However, I think it is more important to tell the influence of large flood on the DOC load. So the daily discharge is a basic standard to judge the "flood". Meanwhile, it is hard to calculate the lasted time of a flood without a minimum standard of a "flood".

L313. These correlations with discharge – were there distinct correlations during each year or during each season? You could carry out an ANCOVA to see whether slopes or intercepts of relationships differed for different periods. Response: ANCOVA analysis on DOC concentrations and spectral indexes was conducted for different years in the revised paper. The results were added in Table 1 and 4 (Page 38, 41) in the revised paper. The discussions were also added in lines 403-410, 544-549 in the revised paper.

L389-392. I don't see how the first and second parts of this sentence are connected. Response: The first sentence was deleted in the revised paper.

L416-425. I find that this discussion on runoff sources and relationship between discharge and DOC concentrations needs to be explicit about what water sources are considered to dominate stream flow during low-flow periods. In boreal catchments Also, this positive relationship between DOC concentration and discharge is opposite what

is generally found in boreal peatland catchments in Scandinavia and North America. What is different? Response: The water source during low-flow period was specified in lines 482-503 in the revised paper. However, it is really hard to explain the reason why there is different relationship with that in North America, for there is no enough data or information. I think it will be done when enough field data were listed together (soil features, seasonal discharge pattern, especially landform and peat locations in the catchment).

L452. I do not agree that peatland-derived DOC should be considered "autochtonuous". Usually that term implies DOC derived from aquatic primary production, and is contrasted to "allochtonous" DOC derived from the terrestrial surroundings. Response: The words "autochtonuous" and "allochtonous" were deleted to avoid misunderstanding in the revised paper.

L511-526. This seems to me like one of the most interesting findings of the study, and could be elaborated on more. Response: More discussion about the finding was added in lines 504-537 in the revised paper.

L538. The word "synthetically" is not appropriate – I believe you mean something like "through synthesis". Response: The word "synthetically" was not appropriate and was deleted in the revised paper. The discussion part on the content was largely re-written.

L555. I don't agree that DOC export among ecosystems within a catchment need to be proportional to the soil organic C stock of each ecosystem. There is no data to support this assumption. Response: The assumption was not appropriate and was deleted in the revised paper.

L563. The term "verified NEE" should not be used. Perhaps you can use the term Net Ecosystem Carbon Balance (NECB), but then you also need to take into account methane-C emissions. Response: Thanks for the suggestion. "Net Ecosystem Carbon Balance (NECB)" is very good. The definition was cited in lines 431-438 in the revised paper. And the methane-C emissions has been accounted in the data as shown in line

429.

L579. This conclusion does not take into account shifts in water quality, including DOC concentrations, due to deepened flow-paths following active layer deepening through the mineral soil under the 30-40 cm organic soil. Response: Thanks for the comments. Theoretically, temperature rise will lead to the deepened flow-paths resulting lower DOC concentrations. However, the expected declining of DOC concentrations would only happen in the low-flow period when only accounting for only a small part of DOC loads for whole year. The annual DOC loads mainly come from the flood periods, and the DOC is mainly come from the upper organic soil. Therefore, the influence of deepened flow-paths on total DOC load should not be decisive when compared to the total stream discharge.

L581. Stating too high precision on projected increases in precipitation. Also, what are the projected changes to evapotranspiration? If ET increases more than precipitation, then we would expected reduced runoff of course. Response: Thanks for the comments. There is really no estimation about the change in evapotranspiration in the region. The emphasis of the study is to make clear the DOC dynamics following the discharge. There is no sufficient data to support my forecast of future DOC loads. Therefore, the third question put forward in the last part of "Introduction" section was deleted in the revised paper.

L616. The definition of NEE only includes the land-atmosphere exchange of $CO_2$ – you have further included DOC export in this NEE estimate that you compare DOC export to, which is not correct. Response: Thanks for the comments. In fact, the data of NEE from the Miao (2014) included the contribution both the $CO_2$ and $CH_4$. This information was added in the revised paper in lines 429.

Fig7 and 8. I would recommend that you indicate at what depth the soil type changes from peat to mineral. Response: Thanks for the comments. The depth shown by dotted line was added in the two figures (Page 49 and 50).

Please also note the supplement to this comment:
https://www.hydrol-earth-syst-sci-discuss.net/hess-2017-412/hess-2017-412-AC2-supplement.zip

---

## Author Comment (AC3) · 31 Oct 2017

The comments from reviewer 3:

Comment 1. DOC is difficult to measure during storms, and the authors have done a great job of capturing the rise and recession of several storms - this in itself is a great contribution. Generally, there is more that could be done with the available data to support the authors' claims and investigate the relationship between discharge and DOC. For instance, plotting discharge vs. DOC could help understand the relationship

between these two and aid in considering a wetter future. Breaking the data set up by years could also be useful, given the large differences in precipitation between the years. There is lots of speculation about the source of the carbon and the flowpaths between the catchment and the stream. I think this speculation significantly detracts from the paper. The authors' analysis relies heavily on several references from other systems, which may or may not be representative of conditions in their system. The authors do a fair job of addressing their second research question regarding the relationship between runoff processes and DOC, but do not – and probably are not capable of rigorously answering their third question regarding the effect of permafrost degradation and climate change given the data set.

Response: Thanks for the comment! First, the relationship between discharge and DOC concentration was plotted in the revise paper. The data set was broken into three years for detailed analysis. Second, the discussion about the flowpaths between and stream was somewhat redundant, and the content was largely cut down in the revised paper. Third, it was indeed no sufficient data to support the third question in the study, and hence the third question was deleted in the context of "Introduction". Therefore, the forecast of DOC loads under changing climate is only a part of auxiliary content of discussions. Finally, several important references and conclusions from similar catchments were cited to give more detailed discussions in the revised paper.

Comment 2. Connectivity vs variable source areas – what is the ultimate cause of the observed trends between discharge and DOC and the fluorescence indices? – the authors argue that the thaw depth controls DOC export concentration and quality, and make some assumptions about contributing areas and catchment connectivity, but with minimal support beyond a few references from other well-known catchments studies. The authors do not have any data from the forested hillslopes, which is an important endmember necessary to substantiate their claims. Especially the paragraph from 428–450 appears highly speculative.

Response: Thanks for the comment! Indeed, we have not measured the DOC from the

upland mountains. Hence, the conjecture bout the hydrological connectivity and DOC source from the upland were deleted in the revised paper according to the comment. The related content may be investigated in the future.

Comment 3: Generally, I found the paper to be very light on analysis - a major finding is that DOC is positively correlated to discharge. It would be nice to see this relationship plotted. Is it linear? Non-linear? Showing this would add further support to the author's claim that this system is transport-limited and that increased rainfall will lead to increased C export. Response: Thanks for the comment! The relationship between discharge and DOC concentration was plotted for each year in the revise paper (Fig. 4, Page 43). There were significant linear relationships.

Comment 4: Consistency in terms – at line 316 – "hydrological DOC, Q, conductivity, and turbidity', earlier at line 288, "discharge turbidity" and "discharge conductivity". Response: Thanks for the comment! The terms were modified in the line 282.

Comment 5: Unclear interpretation of FI index – FI varies in the soils from 1.3 to 1.55 and in the stream from 1.43 to 1.62. The authors assume that the range indicates ". . .both terrestrial and microbial sources" (line 320), and cite Cory's 2010 paper which focuses on correcting fluorescence spectra from different instruments. This is not an appropriate reference to support the authors' interpretation. McKnight's 2001 publication in which microbial and terrestrial end members is defined would be a better choice, but still it would be useful to present more rationale for the authors' interpretation, and to address alternate hypotheses to explain the differences – like the influence of distal water sources.

Response: Thanks for the comment! The reference of McKnight (2001) was well studied and cited in the revised paper in line 213.

Comment 6: Stable isotopes – the authors argue that stable isotopes indicate that peat porewaters, rather than direct rainfall or mineral soils are the source of runoff. But they have not measured isotopes from the mineral soils beneath the peats, or from the

more distal parts of the catchment (ie. the hillslopes), and thus the argument is weak. Furthermore, it is unclear from Figure 6 whether they've collected enough samples to see changes to the isotopic composition for individual storms, which limits the potential inference. Unclear how total DOC export magnitudes were estimated – there are no methods regarding the calculations used.

Response: Thanks for the comment! First, the isotope data of soil pore water were from the whole active layer which include the lower mineral soil in summer. Meanwhile, the isotope data from hillslopes in 2013 were added in Fig. xx in the revised paper according the comment. Second, the DOC loads were re-estimated by a new method according to the suggestion from the first reviewer. The DOC load was re-calculated by the program LOADEST with the web-based calculation program (https://engineering.purdue.edu/mapsever/ldc/LOADEST, version 2012). The new DOC yield estimated by the program is 4.7 g m-2 yr-1. The new result will used in the revised paper (In lines 218-232, line 302).

Comment 7: (miner) 56 – missing some major references regarding the effects of permafrost thaw on hydrology, for instance Hinzman et al., 2005, Jorgenson et al., 2006... Response: Thanks very much for the reference list provided for me! Some important references were collected and cited in the revised paper.

101 – 103: How do you define 'satisfactory'? Many studies have focused on the fate of permafrost carbon – Drake et al., 2015 is a good example and much of the BDOC loss methods by Wickland and others. Spencer et al., 2015 provides a nice conceptualization of the fate of permafrost carbon.

Response: Thanks for the references. There is indeed some studies on the fate of permafrost carbon including DOC. The sentence was modified in the line 79-84. The reference of Spencer et al., 2015 was added in line 81.

128 – This is very broad question involving multiple disciplines and I'm not convinced that your data set is nearly enough to address this. I would suggest removing this

question all together. Response: The question was removed in the revised paper.

161: What is 'yang'? Response: It was replaced by "young".

177: How soon is ': : :as soon as possible.'? Hours? Days? Weeks? This matters especially when you're talking about DOC. Were samples stored in a cool, dark place before analysis? Response: The detailed procedures to storing water samples was added in lines 151.

183 and 190 – sensors are not consistently or properly referenced (ie. Campbell, USA and YSI6600, USA are both incomplete) Response: The detailed information for the instruments were added in lines176 and 184.

203: Missing a verb – maybe 'collected'? Response: The verb was added: "rainfall samples were collected during the two growing seasons."

271: This sentence is confusing. Do you mean that there was no standing water in the peat? Response: The sentence was re-written: "No water level higher than peat surface were detected for the three years." Line 264 in the revised paper.

288: I think you can remove the word 'discharge' and just say 'turbidity'. Similarly, 'electrical conductivity' is clearer than 'discharge conductivity'. Response: The word "discharge" was removed. (Line 282 in the revised paper)

319: Is it reasonable to assume that the FI range will be similar in your system to those studied by Cory? Response: The reference was replaced by McKnight (2001) according to the comment.

347: If they were not statistically different, wouldn't the p value by larger than the 0.01 test threshold? Response: The p values should be larger than 0.01. The error was modified in the paper. (Line 354 in the revised paper)

389: This sentence is not clear, and not totally true. Response: The whole sentence was removed in the revised paper.

397: This statement may be generally true, but not always. Organic soil macropores may not exist everywhere. Do they exist in the Fukuqi catchment? The high hydraulic conductivity and porosity of shallow soils relative to deeper soils also plays an important role. Response: Thanks for the comment! The information about the porosity of the upper organic soil is listed in line 125 in the revised paper. The higher hydraulic conductivity of shallow organic soil compared to lower mineral soil is discussed in Section 4.3 in line 493-504 in the revised paper.

407: This may be evidence, I don't think that it's necessarily proof. Response: The sentence was removed in the revised paper.

416: What do you mean by 'fundamental condition'? Response: I have meant that lateral subsurface flow was an important condition of the positive relationship. The word "fundamental" was replaced by "important" in the revised paper.

418 – 420: I do not believe that subsurface flow "guarantees" that water closest to the stream will always reach the stream first. When subsurface conditions are homogeneous, this may be true, however soil pipes in organic soils (Carey and Woo 2001) and mineral soils (Koch et al., 2013), tussocks (Quinton et al., 2000), and hetereogeneity in subsurface soils (Koch et al. 2017; Laine-Kaulio 2014 and 2015) may complicate this and lead to preferential areas of flow, allowing some areas further from the stream to contribute faster and more than areas near the stream. Response: It is true that high soil hetereogeneity may lead to preferential flow in some region. But in fact, the miner soil under the peat was very uniform without large soil pipes. The discussion is only a conjecture and not an important content in our study. The related sentences were re-written in lines 493-496.

420: I don't quite follow this sentence. Maybe break it down into a few sentences? Also, the positive correlation between Q and DOC is likely a result of more dynamics than simply the proximity of organic-rich soils, it also implies that the source is large, and that the presence in the stream is transport-limited. This point seems important

to the story you're telling. Response: Thanks for the comment! The sentences were re-written in the revise paper. The idea of "transport-limited" is important for the study. The conclusion was expressed in the Section 4.2 in lines 447-456 in the revised paper.

428: Spence and Woo 2006 and Spence and Phillips 2015 both support this point and provide useful precedent. Response: Thanks for the comment! However, the discussion about the hydrological connectivity is deleted in the revised paper due to the lacking of convincing support by enough field data. 431: "Geomorphic landscape structures" is kind of vague. Response: The sentence was removed.

436: This sentence is unclear – if the peatland is highly conductive, shouldn't it facilitate movement of water from the hills? Response: The discussion about the hydrological connectivity is deleted in the revised paper due to the lacking of convincing support by enough field data.

441: I don't think you can assume the values of the hillslopes. I would not expect them to look more like rain than the peatland soil porewaters. Response: Thanks for the comment. The discussion about the hydrological connectivity is deleted according the comment.

484: Where is the evidence that these don't generally change with DOC concentration? Response: This is not an appropriate opinion, and the sentence was re-written in lines 486-489 in the revised paper.

451: I believe that your data very much supports an allochthonous DOC source – autochthonous DOC would result from in-stream processes like the degradation of photosynthetic cells. Variations in contributing area also likely play an important role. Response: Two words "allochthonous" and "allochthonous" were removed to avoid misunderstanding.

505 – 510: Koch et al. 2014 found that stream chemistry changed much earlier, around mid-June concurrent with the beginning of thawing of the mineral soils. Based on your

depth to ice measurements and statement that the organic-mineral boundary is near 30 – 40 cm, it seems like you should also start seeing this response somewhere in June. Autumn sounds too late – by this time you've reached maximum thaw and in fact may be beginning to freeze again.

Response: The data around the middle June was considered when thaw depth reaching the mineral soil. In 2014, we found the FI and BIX values increase after June without the disturbance of rainfalls. The trend was discussed in lines 513-525 in the revised paper. However, no exact beginning point can be identified in our study.

536: I don't understand the logic here: how can you suggest that HIX values are not sensitive to soil active layer depths when you show substantial variations in HIX with soil depth (Figure 8)? Response: The conclusion was incorrect and was removed in the revised paper.

555: There are two assumptions here and I'm not sure if either is reasonable: 1. Is it reasonable to assume that export is proportional to concentration for both forest and peatland systems? Forest and peatland carbon is fairly different, and I imagine could have differing levels of leachability and solubility, and thus transport potential. And I don't believe that you've discussed forests at all before this point. 2. This seems to ignore your previous claims that only the peatland contributes to the stream DOC pool. Response: Thanks for the comment! The estimation for DOC export from peatland was removed as having no enough data to support the assumptions.

587 But at the same time temperatures are likely to warm, impacting overall carbon stocks and DOC production. So there are lots of variables that will likely affect the active carbon pool. Response: It is really difficult to forecast DOC export in the conditions of temperature rise and rainfall change. However, there is the largest possibility that DOC export increases with rising rainfall, as the DOC export process is "transport-limited" but not "source-limited".

Figure 1 needs lat/longs Response: The information was added in Fig. 1. (Page 42)

Fig 2 – Date format is difficult to read and in strange increments. What does 'standing water level' mean? Is this level and the thaw depth from one point? How representative is this point? Is this point shown in Figure 1? Response: The date format was modified. The "standing water level" was changed into "Water level". The information about the water level gauging point was in the section 2.2. "Sampling and monitoring program". The information was also added in Fig. 1 in the revised paper.

Fig 6 – It would be nice to also have discharge on this plot to see how stream water isotopes relate to discharge. Response: The discharge data was added to the plot in Fig. 7 in the revised paper (Page 48).

Fig 7 – Probably don't need negatives on the y axis – What would a negative soil depth mean? Why not set up this plot like those in Figure 8? It would make it easier to compare the seasonal trends. Response: The soil depth was modified to positive value in the revise paper (Page 49, 50). However, if put the data into one figure like Fig. 8, it is too complex to identify the lines.

Please also note the supplement to this comment:
https://www.hydrol-earth-syst-sci-discuss.net/hess-2017-412/hess-2017-412-AC3-supplement.zip

---

## Author Response (AR1)

**Response to the editor**

Thanks for the comments from the editor summarized on 15[th] Nov 2017. According to the comments, the paper was revised carefully. A number of technical problems were modified, and the manuscript was edited by a native English speaker from U.S.A. The responses to the comments and some main modifications were listed as follows:

Comment 1. The use of stable isotopes does not support the manuscript. At no point do you explain why there is a large offset between rainfall and the stream/soil water samples (I'm assuming snowmelt recharge?). The congruence between stream and near-stream values does not supported your response to reviewer 3 and reference to a figure (xx) exists. I suggest that the entire isotope analysis can be removed from the manuscript to more closely pursue the DOC quality story.

Response: Thanks for the comment! It is fact that we have no enough data to explain the large offset between rainfall and stream/soil water samples, as well as the congruence between stream and soil pore water. Therefore we accepted the advice and removed entire isotope analysis.

Comment 2. You have addressed the inter-annual variability and applied LOADEST. The link you have provided has changed

(https://engineering.purdue.edu/mapserve/LOADEST/). Regardless, there is little explanation as to why there is this inter-annual variability. Later in the conclusion you state that there is a stable relationship between DOC and discharge - which is clearly not the case. More detail into the residuals and variability of the Q-DOC relationship is warranted because you lean very heavily on the premise that DOC is transport limited, but the relationship is not particularly stable year-over-year.

Response: The URL was updated by the right one as listed above in the revised paper (Line 225). The description about the variation in DOC load estimated by LOADEST was supplied in the revise paper (Lines 303-314).

The original viewpoint of "a stable relationship" was a wrong expression, and therefore was removed. Meanwhile, the inter-annual variation in the relationship of Q-DOC was also discussed about the controlling factors (Lines 553-574).

Comment 3. The issue of the riparian zone and its importance is not supported by data in this manuscript and should be revised (lines 473-481). There is not sufficient spatially explicit field data to resolve this level of process detail.

Response: Thanks for the comment! The content about the riparian zone

was deleted in the revised paper. The DOC source was discussed only in the section 4.2 (Lines 443-510). The DOC export capacity was discussed only in the section 4.3 (Lines 543-552).

Comment 4. I have some issue regarding the system is not DOC limited based on the uncorrelated DOC vs temperature in the soils. The only way this could be evaluated (and it should be referenced) is with more data for a single depth-location vs temperature as opposed to pooled data.

Response: Thanks for the comment! In the original paper, it was indeed incorrect to say "the system is not DOC limited" because no enough data supporting the idea. However, the really important thing was to highlight "the system is transport limited". There is no need to discuss why "the system is DOC limited" because it deviated from the main idea. Therefore, the related content was removed in the revised paper. Only the discussion about the large DOC productive potential in the peatland was added in the revised paper. (Lines 543-552 )

Comment 5. It is noted that the original manuscript was very similar to the Guo et al. (2015) paper in the Journal of Hydrology. I would warn the lead author against self plagarization as many of the sentences were virtually the same.

Response: Thanks for the comment! I know the paper is similar to another published paper in some aspects. But there are some major differences between the two papers including the study region, estimation method, statistical method and sampling extents, and so on. Importantly, we make clear the relationship between flowpath-shift and DOC chemical characteristics in this paper, and give a detail statement on the DOC load and inter-annual variation. In short, we try to avoid self-repetition as could as possible.